# Dissecting Cellular Mechanisms of Long-Chain Acylcarnitines-Driven Cardiotoxicity: Disturbance of Calcium Homeostasis, Activation of Ca^2+^-Dependent Phospholipases, and Mitochondrial Energetics Collapse

**DOI:** 10.3390/ijms21207461

**Published:** 2020-10-10

**Authors:** Alexey V. Berezhnov, Evgeniya I. Fedotova, Miroslav N. Nenov, Vitaly A. Kasymov, Oleg Yu. Pimenov, Vladimir V. Dynnik

**Affiliations:** 1Institute of Theoretical and Experimental Biophysics, Russian Academy of Sciences, Pushchino 142290, Russia; alexbereg56@gmail.com (A.V.B.); miroslav.nenov@temple.edu (M.N.N.); polegiteb@gmail.com (O.Y.P.); 2Institute of Cell Biophysics of the Russian Academy of Sciences, Pushchino 142290, Russia; delf-fenka@rambler.ru; 3Alzheimer’s Center at Temple, Lewis Katz School of Medicine, Temple University, Philadelphia, PA 19140, USA; 4Center for Brain Research, Department of Molecular Neuroscience, Medical University of Vienna, 1090 Vienna, Austria; vit.kasymov@gmail.com

**Keywords:** cardiomyocytes, palmitoylcarnitine toxicity, Ca^2+^ overload, L-type calcium channels, Krebs cycle, mitochondrial permeability transition pore, phospholipases

## Abstract

Long-chain acylcarnitines (LCAC) are implicated in ischemia-reperfusion (I/R)-induced myocardial injury and mitochondrial dysfunction. Yet, molecular mechanisms underlying involvement of LCAC in cardiac injury are not sufficiently studied. It is known that in cardiomyocytes, palmitoylcarnitine (PC) can induce cytosolic Ca^2+^ accumulation, implicating L-type calcium channels, Na^+^/Ca^2+^ exchanger, and Ca^2+^-release from sarcoplasmic reticulum (SR). Alternatively, PC can evoke dissipation of mitochondrial potential (ΔΨ_m_) and mitochondrial permeability transition pore (mPTP). Here, to dissect the complex nature of PC action on Ca^2+^ homeostasis and oxidative phosphorylation (OXPHOS) in cardiomyocytes and mitochondria, the methods of fluorescent microscopy, perforated path-clamp, and mitochondrial assays were used. We found that LCAC in dose-dependent manner can evoke Ca^2+^-sparks and oscillations, long-living Ca^2+^ enriched microdomains, and, finally, Ca^2+^ overload leading to hypercontracture and cardiomyocyte death. Collectively, PC-driven cardiotoxicity involves: (I) redistribution of Ca^2+^ from SR to mitochondria with minimal contribution of external calcium influx; (II) irreversible inhibition of Krebs cycle and OXPHOS underlying limited mitochondrial Ca^2+^ buffering; (III) induction of mPTP reinforced by PC-calcium interplay; (IV) activation of Ca^2+^-dependent phospholipases cPLA2 and PLC. Based on the inhibitory analysis we may suggest that simultaneous inhibition of both phospholipases could be an effective strategy for protection against PC-mediated toxicity in cardiomyocytes.

## 1. Introduction

The distinct mechanisms of cardiac dysfunction after acute ischemia-reperfusion (I/R) are not sufficiently studied yet. To date, only a few molecular mechanisms of cardiomyocytes (CM) death in I/R have been proposed. It was suggested that the flux of Na^+^ from one cell to another may result in reverse mode operation of Na^+^/Ca^2+^ exchanger (NCX), intracellular Ca^2+^ ([Ca^2+^]_i_) accumulation, cell-to-cell propagation of hypercontracture, and CM death [1]. Besides, it was shown in [2] that palmitoylcarnitine (PC), known ischemic metabolite, induced Ca^2+^ overload, implicating L-type calcium channels (LCC) and Ca^2+^ release from the sarcoplasmic/endoplasmic reticulum (SR/ER) stores. At this time it was determined that long-chain fatty acids (LCFA) or respective acylcarnitines (LCAC) may activate or inhibit LCC [3,4,5] and activate SR/ER calcium channels [6,7,8]. A few years later, it was recognized that limited mitochondrial Ca^2+^ buffering capacity and mitochondrial permeability transition pore (mPTP) induction may underlie the deterioration of cellular Ca^2+^ homeostasis in CM in I/R [9,10,11]. The role of SR/mitochondrial interplay in the control of Ca^2+^ homeostasis at I/R was discussed [11]. The experiments on isolated CM have indicated that, besides Ca^2+^ excess, mPTP opening may be evoked by long-chain fatty acids or acylcarnitines [8,10] which are known to accumulate during I/R [12,13,14,15]. At 10 µM, PC has been shown to induce the dissipation of mitochondrial potential (ΔΨ_m_) and mPTP opening [10]. Recent studies have suggested that PC and Ca^2+^ can be used as mPTP inducers in liver mitochondria, and could act synergistically by reinforcing the effects of each other [16].

LCFA, and especially respective Acyl-CoAs, have multiple actions on cellular metabolic pathways, including: uncoupling of oxidative phosphorylation [17,18], inhibition of oxidative phosphorylation [19,20], inhibition of various NADH-dehydrogenases [21,22,23], inhibition of ADP/ATP and dicarboxylate carriers [24], and induction of oxidative stress and reactive oxygen species (ROS) accumulation [17,18]. The mechanisms providing mitochondrial respiration inhibition, dissipation of ΔΨm, and mPTP induction by long-chain fatty acids or acylcarnitines are not studied in detail yet. It was assumed that harmful effects of LCAC may be related to the direct inhibition of the respiratory chain proteins by long-chain Acyl-CoAs [19,20].

Previous studies indicated inconsistent effects of LCFA/LCAC on Ca^2+^ homeostasis in cardiac myocytes [10,25]. Registered effects were depending on the fatty acid nature. Free myristic and palmitic acids and short-chain acylcarnitines (C6–C12) at the concentrations up to 300–500 µM had a minor effect on [Ca^2+^]_i_ within 30–45 min [25], while long-chain palmitoylcarnitine (PC) and myristoylcarnitine (MC) triggered the spectrum of dynamic regimes depending on the concentrations used [26].

As early as 1999, it was shown that addition of PC evoked immediate increase in [Ca^2+^]_i_ in dose-dependent manner (5–20 µM) [2]. This Ca^2+^ overload was reversible without having any impact on CM viability. However, opposite to these results, subsequent studies have indicated that low concentrations of PC (2–10 µM) increased Ca^2+^-sparks frequency [8,26], evoked single [8] or repetitive [26] Ca^2+^-waves, and triggered steady state dynamic regimes with high Ca^2+^ content in isolated microdomains of CM [26], i.e., regimes characterizing the behavior of the system with positive feedback (calcium induced calcium release from SR). Based on our previous observations, we suggested that activation of Ca^2+^-dependent phospholipases (interplay of PLA2, PLC, and PLD, i.e., “phospholipases vicious cycle”) in Ca^2+^-enriched microdomains may provide membrane phospholipids hydrolysis and induction of plasma membrane nonspecific permeability to cations [26].

In contrast to [2], steep elevation of [Ca^2+^]_i_ registered at high PC or MC concentrations was observed only after a lag period of 2 to 8 min [25]. This lag period was inversely dose-dependent and was absent in CM with suppressed mitochondrial respiration, indicating the implication of mitochondrial Ca^2+^ buffering in this phenomenon.

Currently, it is unclear whether LCC, or transient receptor potential channels (TRPC), involved in store-operated calcium entry (SOCE) [27], may contribute to Ca^2+^ overload induced by LCFA/LCAC excess. Besides, the mechanisms providing the inhibition of oxidative phosphorylation (OXPHOS) and tricarboxylic acid cycle (TCA cycle) by long-chain fatty acids are not studied in detail. The relative contribution of phospholipases and/or mPTP opening in the deterioration of CM Ca^2+^ homeostasis in I/R also require further investigation.

In the present study, we consider LCAC (respective Acyl-CoAs) as an initial trigger that may evoke the redistribution of Ca^2+^ from SR to mitochondria, inhibit TCA cycle and oxidative phosphorylation, limit mitochondrial calcium retention capacity (CRC), and, in combination with calcium, induce mPTP opening. As for the impact of phospholipases activation, arachidonic acid produced by PLA2 may contribute to deleterious effects of exogenous LCFA/LCAC.

## 2. Results

### 2.1. Versatile Dose-Dependent Effects of Saturated Long-Chain Acylcarnitines on Calcium Homeostasis

#### 2.1.1. Sparks and Microdomains with Elevated Ca^2+^ Level

Figure 1A shows that myristoyl-L-carnitine (MC) may evoke Ca^2+^-sparks, and steady state regimes in dose-dependent manner. At low concentrations (3–5 µM), MC transformed irregular Ca^2+^-sparks to almost periodic, increased sparks frequency, or induced high frequency sparks in previously silent areas of CM. Initial Ca^2+^-sparks frequency varied within the limits of 0.1–1 events/s. With time, sparks frequency raised two- to threefold, and the areas occupied with high frequency sparks substantially enlarged (Figure 1A, *t* = 60–63 s and *t* = 85–88 s). It is believed that Ca^2+^-sparks may underlie both the initiation and propagation of slow Ca^2+^-waves [28,29].

Besides that, MC triggered new steady state regimes with high [Ca^2+^]_i_ content in isolated microdomains, marked by arrows in Figure 1A. These Ca^2+^-enriched microdomains appeared at the concentrations of MC equal or higher than 5 µM, and coincided with the regions of high frequency sparks, i.e., spontaneous events of Ca^2+^-release by the clusters of SR Ca^2+^-release channels. The first short living microdomain appeared at *t* = 60 s and than at *t* = 63 s. After *t* = 86 s, it transformed to a long living microdomain (right part of Figure 1A).

The appearance of Ca^2+^-enriched long living microdomains demonstrates the switching of local calcium signaling machinery (of the clusters of ryanodine receptors (RyRs)) from quasiperiodic to steady state regime, having a lifetime from several seconds to several minutes. Here, we may suggest that such microdomains may be formed at the regions with the lack of close contacts of SR Ca^2+^-release channels and mitochondrial Ca^2+^ uptake system.

Application of PC (3–20 µM) evoked similar dynamic regimes, while short-chain acylcarnitines (C6–C12, up to 100 µM) had little effect on [Ca^2+^]_i_ within 30–45 min of recordings [25,26].

#### 2.1.2. Delayed Ca^2+^ Overload, Hypercontracture, Nonspecific Cationic Permeability, and CM Death

At high concentrations of 20 to 50 µM, PC and MC induced Ca^2+^ overload, hypercontracture, and CM death (Figure 1B). Threshold concentration of PC evoking Ca^2+^ overload (PC*) was equal to or above 20 µM. Ca^2+^ overload was observed only after a lag period of 2 to 8 min. The duration of the lag period was inversely dose dependent. Steady state regimes, observed in Ca^2+^ enriched microdomains, could have appeared soon after PC load, and existed till the end of the lag period.

Figure 1B demonstrates that 20 µM PC evokes: Ca^2+^ overload after lag period of 210 s (bright rod shape cell at *t* = 210 s), hypercontracture and rupture of plasma membrane characterized by fluorescent dye loss (bright oval cell at = 220 s), and final cell death (red oval cell, *t* = 365). The examples of respective changes in cytosolic [Ca^2+^]_i_ and [Na^+^]_i_ concentrations, registered throughout the time after the application of PC, are presented in Figure 1C. Time periods of cell shortening are marked as red hatched boxes in panel C. Here, steep elevation of [Ca^2+^]_i_ (red lines, left) at the onset of Ca^2+^ overload is accompanied by hypercontracture (boxes), while steep rise of [Na^+^]_i_ follows it (blue lines, right). The delay in [Na^+^]_i_ rise indicates that internal mechanisms may underlie the induction of hypercontracture [2], while the induction of nonspecific cationic permeability by LCFA excess [30] may not realize in the conditions of our experiments.

### 2.2. Ca^2+^-Overload Does Not Depend on External Ca^2+^ Influx

In 1999, it was shown that PC (5–20 µM) induced immediate dose-dependent increase in [Ca^2+^]_i_, which was reversible and not accompanied with CM death [2]. Removal of Ca^2+^ from extracellular media ([Ca^2+^]_o_ = 0) attenuated the rate of [Ca^2+^]_i_ increase (the slope) and slightly diminished the peak [Ca^2+^]_i_. Meanwhile, the blockade of LCC and NCX affected both parameters, diminished the slope and peak [Ca^2+^]_i_, indicating that the implication of Ca^2+^ influx pathways in the elevation of Ca^2+^ at PC excess. To test whether PC-induced Ca^2+^ overload and CM death depend on an influx of Ca^2+^ from the extracellular media, the effect of PC was investigated at different values of [Ca^2+^]_o_ and in the presence of the blockers of Ca^2+^ influx channels.

#### 2.2.1. Ca^2+^-Overload and Myocytes Death Is Observed in Ca^2+^-Free Medium

Figure 2A shows the effects of [Ca^2+^]_o_ variation on the parameters characterizing Ca^2+^ overload induced by PC. Representative traces are depicted in the panel. Here, omitting extracellular Ca^2+^ for 2 min ([Ca^2+^]_o_ = 0) resulted in a substantial (about threefold) fall in peak [Ca^2+^]_i_ (red vs. black line). However, even this moderate elevation in [Ca^2+^]_i_, observed in Ca^2+^-free media, was sufficient to produce CM hypercontracture and death. Meanwhile, exclusion of extracellular Ca^2+^ for 20 min abrogated Ca^2+^ overload and cell death, apparently, by preventing refilling of SR Ca^2+^ stores over this long period of time (green line).

Collectively, these results may indicate that the intracellular stores (SR and mitochondria) may have enough calcium to induce Ca^2+^ overload in CM affected by PC. Indeed, there is evidence that the calcium released from intracellular stores (SR/ER) is not sufficient to induce mPTP opening, and corresponds to about 50% of the total Ca^2+^ pool required for pore opening by Ca^2+^ excess [31]. However, at this point, we can speculate that PC may be considered as factor X, which, in combination with Ca^2+^, induces Ca^2+^ overload and CM death. Here, minimal concentration of PC (PC*), evoking Ca^2+^ overload, is equal to or above 20 µM.

#### 2.2.2. L-Type Calcium and Transient Receptor Potential Channels Do Not Significantly Contribute to Ca^2+^ Overload

Presently, it is unclear whether LCC and TRPC1,4,5, involved in SOCE, may contribute to Ca^2+^ overload induced by PC or MC excess. However, the results of 1999 indicated that Ca^2+^ entry via LCC may be involved in the increase in peak [Ca^2+^]_i_ evoked by PC [2].

Figure 2B describes the effect of L-type calcium current blocker nifedipine on 40 µM MC-induced Ca^2+^ overload. Average values of [Ca^2+^]_i_ (as Fluo-4 ΔF/F_0_) are depicted in pane B for all cells (*n* = 7, 8). In the figure, the inhibition of L-type calcium current with nifedipine not significantly affected the lag period, slope, and peak [Ca^2+^]_i_ in comparison with the control values (trace 2 vs. 1).

Similarly, TRPC4,5 blocker ML-204 not substantially altered the parameters of Ca^2+^ overload in most of the experiments. However, we have found that blockade of TRPC4,5 abrogated fast Ca^2+^ oscillations triggered by MC in some cardiomyocytes. These rare regimes with high amplitude fast Ca^2+^ oscillations were observed only in 10 to 15% of all experiments. It may be 2–3 of 7–10 cells in a coverslip, which are resistant to PC toxicity and contract in spite of high average Ca^2+^ level. Figure 2C,D depict representative control and typical Ca^2+^ traces of the effect of TRPC4,5 blocker ML-204 on MC-evoked Ca^2+^ responses in such groups of cells. Viability score (VS), which is the ratio of viable to total cells in the observed area, is indicated by corresponding values in panels C and D. Here, we consider the viability as the potency of contracted cell to preserve membrane integrity, and plasma and mitochondria membrane potentials.

In the control experiments, seven of nine cells responded with typical steep Ca^2+^ rise, which was accompanied with hypercontracture and death within 5 to 6 min after addition of 40 µM MC (representative red trace at panel C). Meanwhile, in two cells, MC triggered fast Ca^2+^ oscillations with a high value of average intracellular Ca^2+^ level (dark blue trace). These oscillations were preserved over period of 30 min in contracted but still viable cells (VS = 2/9). Panel D shows that the inhibition of TRPC4,5 did not have any significant effect on the values of lag period, slope and amplitude of Ca^2+^ rise in 5 dying cells (compared to control cells of panel C), but prevented Ca^2+^ oscillations in the rest 3 cells that remain viable within 30 min (dark blue representative traces, VS = 3/8).

Collectively, these results indicate minor involvement of both types of channels in PC-evoked Ca^2+^ overload and CM death. As for Ca^2+^ oscillations, this mode of CM response to PC may strongly depend on the value of Ca^2+^ influx in SR stores, which may be affected by PC and abrogated by TRPC4,5 blockers.

#### 2.2.3. The Effects of PC/MC on L-Type Calcium Current in Patch-Clamp Experiments

Currently, the data presented above are in opposition to earlier results of others. Previous reports have demonstrated that PC may strongly activate [3,4] and inhibit [4,5] L-type Ca^2+^ current (Ica^2+^). The experiments performed on LCC, incorporated in the lipid bilayer, revealed strong dose-dependent and time-dependent effects of PC on open probability of the channels, with maximal effect achieved at 1 µM PC and within the first minutes of the recording [4]. Although, according to the results of the experiments on isolated CM, 10 µM PC exerted immediate inhibition of Ica^2+^ [5].

To address the problem, we investigated the effect of PC on Ica^2+^ in a whole cell perforated patch-clamp experiment using acutely isolated cardiomyocytes from rat heart left ventricle. The results presented in Figure 3 demonstrate dose-dependent and time-dependent influence of PC on Ica^2+^. In our study, PC induces either a potentiating or a biphasic effect on cardiomyocyte Ica^2+^ (panels A, B): (I) with transient Ica^2+^ activation, which reached the maximum starting within 2 to 3 min in cardiomyocytes of group 1 (green squares, Figure 3A); (II) initial activation with subsequent Ica^2+^ inhibition in late phase for group 2 of cardiomyocytes (red squares, Figure 3). Importantly, CM can be statistically divided into two groups based on their responses to PC (Figure 3C). Further analysis revealed significant correlation between the levels of initial PC induced Ica^2+^ potentiation and subsequent PC mediated Ica^2+^ inhibition. As can be seen from Figure 3D, cardiomyocytes can be separated into two different groups based on the correlation showing that the higher the PC-induced potentiation of Ica^2+^ current, the lower the subsequent inhibition. Similar results were obtained for MC effects on Ica^2+^ (data not shown).

The first group of cells (about 44% of all), i.e., initial phase cells, displayed substantial activation of Ica^2+^ by PC, without visible suppression of Ica^2+^ in the late phase of responses (green trace in panel A). Maximum activation of Ica^2+^ achieved 18% of control value 5 min after the application of 20 µM PC (green boxes in panel C). Meanwhile, in the second group of cells (56% of all), in the late phase cells, 20 µM PC evoked significant reduction in Ica^2+^ (to 40% of control Ica^2+^) without of its marked initial activation (red trace in panels B, red box at D).

Addition of 20 µM MC induced approximately two times lower effects (activation and inhibition of Ica^2+^) in both groups of cells compared to the effects of 20 µM PC (not shown). Application of 50 µM PC or MC resulted in abrupt Ica^2+^ “rundown” accompanied by hypercontracture and CM death (brown triangles in panels E and F, respectively) which might be associated with a detergent effect of high doses of LCAC. Collectively, we can suggest that weak activation of Ica^2+^ in the first group of cells, and feedback inhibition of Ica^2+^ by cytosolic [Ca^2+^]_i_ in the second group of cells, underlies a weak impact of LCC on Ca^2+^ overload induced by PC (MC).

### 2.3. Sarcoplasmic Reticulum (SR)—Mitochondria Interplay

Previous studies have indicated that close localization between SR and mitochondria may create functional interplay among them in limited microdomains with the exchange of Ca^2+^ [32,33,34], providing rapid Ca^2+^ buffering in mitochondria and limiting [Ca^2+^]_i_ [33]. As for pathophysiological relevance, it has been hypothesized that the interplay between SR and mitochondria may be involved in the induction of aberrant Ca^2+^-sparks, propagation of Ca^2+^-waves [29], and may increase the susceptibility for mPTP in the areas of close contacts of mitochondria with SR [35,36,37,38].

Palmitoyl-CoA is known to activate SR Ca^2+^ release channels, apparently by increasing the dissociation of FKBP12.6 from RyR [39]. Besides, PC (10 µM) induces dissipation of ΔΨ_m_, CM mitochondria swelling, ROS overproduction [10], and increases RyR oxidation and SR Ca^2+^ leak [8]. However, the effect of PC on SR-mitochondria interplay requires further investigation.

#### 2.3.1. Redistribution of Calcium from SR to Mitochondria Evoked by PC

Figure 4 demonstrates that PC evoked the redistribution of Ca^2+^ from SR (black line) to mitochondria (blue line). This effect may be observed even at low concentrations of PC (2–5 µM) that cannot induce visible alterations in [Ca^2+^]_i_ and Ca^2+^ overload. Addition of 5µM PC immediately activated Ca^2+^ release from SR that was manifested by the decrease in SR Ca^2+^ content ([Ca^2+^]_SR_, black line) and reciprocal rise in mitochondrial Ca^2+^ ([Ca^2+^]_m_, blue line).

The suppression of SR Ca^2+^-cycling by the application of thapsigargin and caffeine further accelerated decrease in Ca^2+^. Thapsigargin is known to inhibit SR Ca^2+^-ATPase (SERCA) while caffeine stimulates release of Ca^2+^ from SR (black line). The uncoupling of oxidative phosphorylation by FCCP evoked the dissipation of ΔΨ_m_ and concomitant steep fall of mitochondrial [Ca^2+^]_m_ (blue line).

Thus, reciprocal alterations in [Ca^2+^]_SR_ and [Ca^2+^]_m_ induced by PC, demonstrate SR-mitochondria interplay in calcium handling.

#### 2.3.2. Mitochondrial Calcium Buffering Underlie the Lag-Period Preceding Steep Ca^2+^ and Na^+^ Rise

Partial dissipation of ΔΨ_m_ by the uncouplers of oxidative phosphorylation or the inhibition of OXPHOS, both these maneuvers diminish mitochondrial Ca^2+^ buffering, the measure of which is calcium retention capacity (CRC) [31,40,41]. CRC is used to evaluate the sensitivity of the mPTP opening in isolated mitochondria and cells. The sensitivity of mPTP to Ca^2+^ load is enhanced by oxidative stress, inorganic phosphate (P_i_) and adenine nucleotide depletion, etc. [40,42,43]. It has been well established that P_i_ promotes mPTP opening but augments protective effect of cyclosporine A (CsA) at Ca^2+^ excess [42,43]. However, whether P_i_ is an mPTP inhibitor or inducer remains uncertain. The experiments of 2003 indicate a dual concentration-dependent effect of Pi on CRC (see Figure 2 in [40]).

Here, we address the effect of CRC variation on Ca^2+^ overload induced by PC. To increase CRC, we varied the concentration of Pi in the medium ([P_i_]_o_). Phosphate facilitates massive Ca^2+^ loading of mitochondrial matrix, by forming the complexes with Ca^2+^ (Ca_3_(PO_4_)_2_), thereby decreasing the concentration of free [Ca^2+^]_m_ and increasing CRC [40].

Figure 4B demonstrates that tenfold rise of [P_i_]_o_ in the medium from 0.4 mM to 4 mM provided threefold increase in the lag period preceding steep rise in [Ca^2+^]_i_, which may indicate the enlargement of mitochondrial Ca^2+^ buffering by P_i_. Incomplete inhibition of mitochondrial Ca^2+^ uptake (of mitochondrial Ca^2+^ uniporter, MCU) with Ruthenium Red (RR) abolished fine mitochondrial Ca^2+^ handling and evoked slow rise of [Ca^2+^]_i_ until the onset of Ca^2+^ overload (Figure 4C, brown line vs. red control) indicating the implication of MCU in SR-mitochondria interplay. Similarly, partial inhibition of SR Ca^2+^ uptake with SERCA inhibitor thapsigargin results in slow rise of [Ca^2+^]_i_ [25], demonstrating second arm of the interplay.

To suppress mitochondrial Ca^2+^ uptake, we also applied the inhibitors of OXPHOS oligomycin and rotenone, known inhibitors of mitochondrial ATP synthase (F0/F1) and complex I of the electron transport chain, respectively. Heart cells pretreated with these inhibitors survive on glycolysis, remain rod-shaped, and preserve their contractile functions in spite of lowered concentration of ATP [44]. Figure 4D shows that the inhibitors of OXPHOS abolished delayed Ca^2+^ overload. In the cells with suppressed mitochondrial energy metabolism, MC induced immediate and sustained concentration-dependent increase in [Ca^2+^]_i_ indicating modulation of SR Ca^2+^ cycling and, especially, of RyR channels activity by MC. Even at low (10 µM) concentration, MC induced steady rise of [Ca^2+^]_i_ (green and red lines in panel D). Fine [Ca^2+^]_i_ buffering, preserved in control experiment till the moment of Ca^2+^ overload (mPTP opening, black line), was abrogated in the cells with suppressed OXPHOS (color lines).

Beside SR arm, PC and MC may have a strong impact on mitochondria. LCFA/LCAC, which accumulate in I/R, represent natural inhibitors of OXPHOS. However, harmful effects of LCAC on oxidative phosphorylation, tricarboxylic acid cycle, and mPTP are not sufficiently studied.

#### 2.3.3. Dual Effects of Long-Chain Acylcarnitines on TCA Cycle, Mitochondrial Potential (ΔΨ_m_), Respiration, and NAD/NADH Redox State

To test whether long-chain acylcarnitines inhibit TCA cycle and OXPHOS, the effects of PC and MC were examined on isolated rat heart mitochondria. Glutamate plus pyruvate or malate plus α-ketoglutarate were used as substrates. Initial steady state respiration was preset by ADP that, in the presence of glucose and hexokinase in the medium, provided a high rate of respiration close to State 3 respiration rate (80–90% of VO_2max_) [16]. Current values of the respiration rate may be evaluated by the slope of polarographic tracks (Figure 5, blue tracks, [O_2_]). As shown in Figure 5, PC and MC display dual concentration-dependent effects on the respiration (blue tracks), mitochondrial membrane potential ΔΨ_m_ (red tracks), and redox state of NAD(P)H (green track, panel C).

Substrate effects of PC and MC. Figure 5A–E demonstrate that PC or MC, being added at a low concentration of 10 µM, stimulate mitochondrial respiration. Entering the mitochondria, LCAC are converted to its CoA ester forms that are oxidized in β-oxidation spiral to final products FADH_2_, NADH and Acetyl-CoA. Further, final oxidation of Acetyl-CoA in TCA cycle provides the influx of extra reducing equivalents (FADH_2_, NADH) to the respiratory chain. Here, added PC switches on β-oxidation and supplies TCA cycle with Acetyl-CoA. Finally, the oxidation of PC in form of Palmitoyl-CoA supports a high rate of electrons flux through the respiratory chain and provides the increase in the respiration rate, ΔΨ_m_, and [NAD(P)H].

Reversible inhibition of the respiration and TCA cycle. Second addition of PC (panel A) or third application of MC (panel E) cause fast fall of ΔΨ_m_, suppression of the respiration, and oxidation of NAD(P)H (panel C). This inhibition is reversible and may be prevented by succinate and β-oxybutyrate (panels A and C), or malate (for glutamate plus pyruvate as substrates, panel E) indicating known inhibition of NAD(P)-linked dehydrogenases by long-chain Acyl-CoAs [21,22,23]. Importantly, added succinate increased NAD(P)H fluorescence (panel C), indicating preserved (intact) reverse flow of electrons through the complex I of respiratory chain. At this point, we may suggest that direct control of TCA cycle and related NAD(P)-linked dehydrogenases by the excess of Acyl-CoAs dominates over supposed inhibition of the respiratory chain proteins.

#### 2.3.4. Irreversible Inhibition of OXPHOS and mPTP Opening. PC and Ca^2+^ Interplay

Further increase in PC resulted in irreversible inhibition of the respiration and dissipation of ΔΨ_m_ (Figure 5, panels B and D). Both the respiration and ΔΨ_m_ cannot be restored by the application of succinate, β-oxobutyrate, and malate specifying collapse of energy metabolism. Added FCCP cannot induce further dissipation of ΔΨ_m_, indicating irreversible effect of PC based on mPTP (panel B). Recently, it was shown that PC excess induced irreversible mitochondrial depolarization, mitochondrial Ca^2+^ loss and swelling, collectively indicating mPTP opening [16].

Critical threshold concentration of PC (PC*), required for this irreversible effect, varied a little with the conditions of the experiments, and may be determined at sequential loading of mitochondria with 10 µM PC. In panel B, the value of PC* is equal to 40 µM. In mitochondria preincubated with 20 µM Ca^2+^, calcium reinforced the harmful effect of PC by lowering PC* to 30 µM (panel D), indicating synergistic action of PC and Ca^2+^. Likewise, PC increased the sensitivity of mPTP to Ca^2+^.

CRC and PC* characterize the sensitivity of mPTP to Ca^2+^ and PC, respectively. Panel F demonstrates the impact of PC and Ca^2+^ on mean values of both regulatory parameters. Here, PC-calcium interplay results in the lowering of both parameters. In mitochondria oxidizing 20 µM PC, the critical average concentration of calcium required for pore opening (i.e., CRC) diminished by 20–22% compared to control (brown bars, panel F). Furthermore, vice versa, in the presence 20 µM Ca^2+^ average threshold concentration of PC* diminished by 35% (green bars, panel F). Similar synergistic action of Ca^2+^ and PC on mPTP was demonstrated earlier on liver mitochondria [16]. This result may help to explain the discrepancy between the minimal concentration of PC required for Ca^2+^ overload in CM (20 µM, Figure 1C) and critical PC* value inducing mPTP opening in isolated mitochondria preincubated in Ca^2+^-free medium (40 µM, panel B).

#### 2.3.5. Impact of PC on ΔΨm in CM

The sensitivity of OXPHOS to PC in isolated CM may vary within the same population of cells, or within different compartments of one cell. Figure 6 demonstrates the variety of ΔΨ_m_ responses to PC in isolated CM.

Added 20 µM PC evokes slow depolarization of ΔΨ_m_ (decrease in ΔΨ_m_, green line) in a selected area of one cell (Figure 6A, bottom cell), which continues to fall till the moment of total cell depolarization induced by the uncoupler of oxidative phosphorylation FCCP. This initial slow depolarization can match to reversible inhibitory effect of second bolus PC observed on isolated mitochondria (Figure 5). Meanwhile, in the second cell (Figure 6A, top cell, red line), PC causes initial hyperpolarization of ΔΨ_m_ (corresponding to the substrate effect of PC in Figure 5), which is followed by abrupt partial depolarization (reversible inhibition of OXPHOS). This partial fall of ΔΨ_m_ is observed in undamaged compartments of contracting cells (the time period of cell shortening is marked as a red rectangle), which may indicate that total dissipation of ΔΨ_m_, mPTP opening (irreversible inhibition of OXPHOS), and Ca^2+^ overload take place in distant compartment of the cell. The recordings of ΔΨm presented in Figure 6B demonstrate the variety of OXPHOS responses to 20 µM MC within nearby areas of the same cell. The existence of spatially distinct subsarcolemmal and interfibrillar localization of cardiac mitochondria [45] may underlie registered heterogeneity of responses to PC or MC.

### 2.4. Implication of Phospholipases and CaMKII in Calcium Overload and Myocytes Death

It is well established that, after I/R, enzymatic activities of Ca^2+^-dependent phospholipases A2 (cPLA2α) and C (PLCβ,γ), and Ca^2+^-independent phospholipase A2 (iPLA2γ) are enhanced, aggravating myocardial recovery [45,46,47,48]. Activation of the Ca^2+^/calmodulin-dependent protein kinase II (CaMKIIδ) is also implicated in myocardial failure in I/R [49,50]. Therefore, the inhibition of the isoforms of phospholipases and CaMKII is known to protect against mPTP opening, necrosis, and apoptosis in I/R injury [46,47,48,49,50,51,52].

PLA2 produces arachidonic acid, which is involved directly or via downstream toxic metabolites in the control of Ca^2+^ homeostasis and mPTP [49,50]. Meanwhile, the Ca^2+^-dependent PLC converts phosphatidylinositol 4,5-bisphosphate to diacylglycerol (DAG) and inositol 1,4,5-trisphosphate (IP3), coagonist of Ca^2+^ at IP3 receptor of SR/ER. Furthermore, CaMKII, being involved in excitation-contraction coupling and in the control of NCX/SR/mitochondria axis at Ca^2+^ overload [51], may activate the phospholipases cPLA2α [53,54] and PLCβ,γ [55,56].

#### 2.4.1. Inhibition of Phospholipases and CaMKII Prevent Ca^2+^ Overload and CM Death

To address the issue on the involvement of cPLA2α, PLCβ,γ, and CaMKII in the control of Ca^2+^ overload and CM death, we apply the inhibitors of these enzymes. Average values of [Ca^2+^]_i_ (Fura-2 340/380 ratio) are depicted in panels A and B of Figure 7, including all dead cells in control experiments (black lines in panels A and B) and all viable cells treated with the inhibitors (colored lines). Here, viability score (VS) may characterize the efficiency of the inhibitors in prevention Ca^2+^ overload and CM death.

Figure 7A shows the effects of aristolochic acid and KN93, potent PLA2 and CaMKII inhibitors, respectively, on Ca^2+^ overload and CM death observed at minimal concentration of PC equal 20 µM. Figure 7B describes the impact of U73122, the inhibitor of PLC, and aristolochic acid at high concentration MC of 50 µM. In control experiments, PC and MC evoked 100% CM death within 15 min (black traces in panels A and B). Addition of aristolochic acid (Arist, 100 µM) markedly increased lag period, diminished the slope and peak [Ca^2+^]_i_ compared to control values (panel A, red vs. black traces). In the control experiment, all eight cells died within 15 min after addition of PC (VS = 0/8), while of seven cells treated with aristolochic acid, four contracted oval shape cells remained viable within 35–40 min (red trace, VS = 4/7). The inhibitor of CaMKII KN93 provided a stronger effect on the parameters of Ca^2+^ overload (blue vs. black traces) but had weaker impact on cell viability (VS = 2/6). Combined application of both inhibitors resulted in further decrease in both, the slope and peak [Ca^2+^]_i_, and provided better viability (dark blue trace, VS = 5/8) indicating additive effect of both inhibitors.

Similarly, the inhibition of PLC with U73122 increased lag period, diminished the slope and peak [Ca^2+^]_i_ (Figure 7B, green vs. black traces) but ensured lower viability (VS = 2/7) compared to the effect of aristolochic acid (red trace, VS = 4/8). However, preincubation of CM with both, U73122 and aristolochic acid, resulted in a strong combined protective effect (dark blue trace, VS = 8/9).

In control cells, PC and MC evoked 100% CM death within 15 min (black traces in panels A and B). Combined inhibition of PLA2 and CaMKII provided 60% efficacy (dark blue trace, VS = 5/8, panel A). Meanwhile, simultaneous inhibition of PLA2 and PLC ensured 90% cell survival over 30–45 min (dark blue trace, VS = 8/9, panel B).

#### 2.4.2. Fast Ca^2+^ Oscillations and CM Survival

In the experiments presented above, PC and MC induced steep Ca^2+^ rise, hypercontracture, and death in all cells, while regular fast Ca^2+^ oscillatory regimes were not observed. Panels C through F describe the experiments with the groups of cells predisposed to Ca^2+^ oscillations and partially resistant to long-chain acylcarnitines excess. Such regimes were registered only in 10% to 15% of all experiments.

In the control experiment, 50 µM MC induced spike-like Ca^2+^ rise and fast death within 3–3.5 min in six of eight cells (representative red trace in panel C), while low amplitude Ca^2+^ oscillations were registered in two contracted round shape cells (blue) indicating heterogeneity of CM populations with respect to their responses to MC.

The inhibition of PLC provided survival in three of nine cells (panel D), while that of PLA2 ensured viability in four of seven cells with the mode of Ca^2+^ oscillations (panel E). Finally, combined application of the inhibitors of both phospholipases provided 100% protection. All cells (eight of eight) responded to MC excess with high amplitude Ca^2+^ oscillations.

## 3. Discussion

In this article, we revisited the role of saturated long-chain acylcarnitines in the control of both cellular Ca^2+^ homeostasis and mitochondrial energy metabolism. Most recent reviews devoted to the studies of reperfusion/injury and cardioprotective mechanisms are focused on harmful effects of calcium and reactive oxygen species (ROS) [57,58,59,60]. The mechanisms of acute lipotoxicity are analyzed separately of Ca^2+^ effects and include the inhibition of mitochondrial oxidative phosphorylation and the accumulation of ROS induced by long-chain fatty acids or respective acylcarnitines [59,60]. It is well established that rapid influx and accumulation of Na^+^ and Ca^2+^ in cytoplasm, which generally occurs in I/R, may lead to the accumulation of [Ca^2+^]_i_ and [Ca^2+^]_m_ and mitochondrial (mPTP) or mitochondrial-independent cell death [57,61]. However, beside Ca^2+^, saturated and unsaturated LCFA/LCAC are accumulated in I/R [12,13,14,15]. According to recent data, serum LCAC represents an independent risk factor of acute myocardial infarction in patients with angina pectoris, etc. [62]. Mitochondria isolated from ischemic areas of myocardium accumulate LCAC and respective long-chain acyl-CoA, which are known to suppress oxidative phosphorylation [15]. Therefore, to address the mechanisms of CM death in I/R, we have to analyze both the impact of LCAC on cellular Ca^2+^ homeostasis and mitochondrial energy metabolism taking into consideration Ca^2+^–LCAC interplay.

### 3.1. Diversity of Dynamic Regimes Induced by PC

Previous studies have demonstrated that 5 to 20 µM PC induced immediate but reversible concentration-dependent steep rise of [Ca^2+^]_i_ [2]. Meanwhile, recent reports have indicated that 10 µM PC evoked Ca^2+^ sparks and single calcium waves [8]. Here, we have shown that Ca^2+^ signaling machinery of CM responded to PC- or MC-like multivariate multi-feedback systems, having a discrete set of dynamic regimes. At low concentrations up to10 µM, MC evoked local events like Ca^2+^ sparks, short- and long-living steady states calcium signals with elevated [Ca^2+^]_i_ in Ca^2+^-enriched microdomains (Figure 1A) and low probability regimes, the regimes of distributed system, periodic Ca^2+^ waves [26]. At high concentrations over the threshold of 20 µM, PC induced global events: invariably triggered Ca^2+^ overload, hypercontracture, and CM death (Figure 1B,C) or low probability high amplitude Ca^2+^ oscillations (Figure 3C). Yet, the determination of the threshold value is approximate. At 12–18 µM, PC may induce Ca^2+^ overload after long time lag period. We assume that in our experiments employing confocal microscopy live imaging, the effect of PC and laser-induced influence on [Ca^2+^]_i_ dynamics may overlap.

### 3.2. mPTP and Ca^2+^ Overload

Although the mechanisms leading to ATP-dependent hypercontracture are not completely understood, recent studies suggest that interplay between SR and mitochondria may underlie Ca^2+^ overload, hypercontracture and CM death providing mitochondrial Ca^2+^ accumulation and mPTP opening [9,11]. As for mPTP, the induction of pore opening by Ca^2+^ excess is thought to be a classic mechanism of cell death in I/R. However, it is well established that LCFA/LCAC, which exert complex effects on mitochondrial energy metabolism, may also promote mPTP opening and necrotic or apoptotic cell death [19]. Studies on permeabilized CM have demonstrated that PC induced the dissipation of ΔΨm and mPTP opening in Ca^2+^ free medium, while CsA prevented pore opening [10]. However, existing data on the protective effect of CsA are inconsistent. In the experiments on liver mitochondria, CsA prevented swelling and mPTP opening when the concentration of PC by not far exceeded some critical level [16]. However, in cardiac mitochondria added CsA was ineffective to abrogate mitochondrial Ca^2+^ efflux and restore the respiration at PC excess [15]. Apparently, besides its direct effect on TCA cycle and oxidative phosphorylation, PC (i.e., respective Acyl-CoA) might have an impact on the elements of mPTP complex distinct of CsA targets like cyclophilin D.

### 3.3. SR-Mitochondria Interplay

The present study has indicated that a 2–5 min lag period, which precedes to PC-induced Ca^2+^ overload (Figure 1C), could result from [Ca^2+^]_i_ fine tuning due to SR-mitochondria interplay. We found that enlargement of mitochondrial CRC with Pi increased the value of the lag period approximately up to three times (Figure 4B), while the inhibition of OXPHOS (CRC = 0) abrogated stabilization of [Ca^2+^]_i_ within this lag period and resulted in immediate smooth rise of [Ca^2+^]_i_ (Figure 4D). The slope of [Ca^2+^]_i_ elevation is concentration-dependent denoting direct impact of MC on SR Ca^2+^ release. Inhibition of LCC and TRPC4,5 did not sufficiently alter the values of lag period, slope, or maximum [Ca^2+^]_i_ (Figure 2), indicating a secondary effect of Ca^2+^ influx in cytosol and SR, respectively. Patch-clamp experiments demonstrated a weak effect of PC on LCC and support this notion (Figure 3). Pre-incubation of CM in Ca^2+^ free medium for 2 min diminished maximum [Ca^2+^]_i_ about three times compared to the control (Figure 2A) However, this lowering of maximum [Ca^2+^]_i_ did not prevent hypercontracture and CM death. This result may denote that intracellular Ca^2+^ stores (SR and mitochondria) have enough calcium to induce, in combination with PC, collapse of energy metabolism, mPTP opening, and Ca^2+^ overload. The value of the lag period, preceding steep rise of [Ca^2+^]_i_, depends on the concentration of Pi in the medium (Figure 4B) indicating minor involvement calpain-dependent mechanisms in CM death.

### 3.4. Fragility of Mitochondrial Energy Metabolism

Similar to its influence on Ca^2+^ homeostasis, LCAC exert a complex effect on CM mitochondrial energy metabolism. Figure 5 shows that PC evokes a tri-modal effect on TCA cycle and OXPHOS in isolated mitochondria as follows: (1) activation at 10 µM (substrate effect), (2) reversible inhibition at 20–30 µM based on the inhibition of NADH dehydrogenases by Acyl-CoAs, and (3) irreversible inhibition at concentrations higher 30 µM resulted from mPTP opening. This range of inhibitory concentrations of PC corresponds to those of [15] and varies in dependence of many factors, including the concentration of Ca^2+^ in the medium. Thus, calcium, which is known to activate key dehydrogenases of TCA cycle and respiratory chain proteins, may aggravate harmful effects of LCAC on the proteins of mPTP complex, and vice versa. Even at low concentrations to 10 µM, PC may induce ROS accumulation [10,15]. However, the reversibility of the inhibition of the TCA cycle and oxidative phosphorylation by PC (Figure 1A,C) indicates that ROS are not involved in these effects.

### 3.5. PC and Ca^2+^ Crosstalk

Recent experiments on liver mitochondria revealed a crosstalk between PC and Ca^2+^ in the control of mPTP [16]. Results presented in Figure 5 demonstrate this effect for heart mitochondria. Figure 5 shows that Ca^2+^ reinforces deleterious effect of PC by lowering critical concentration of PC (PC*) required for irreversible dissipation of ΔΨm, from 40 to 30 µM (panels D vs. B). Likewise, PC may diminish the value of CRC. CRC and PC* characterize the sensitivity of mPTP to Ca^2+^ and PC, respectively. Figure 5F demonstrates that, in mitochondria oxidizing 20 µM PC, the average value of CRC diminished by 20–22% compared to the control (brown bars). Furthermore, vice versa, in the presence 20 µM Ca^2+^, the average PC* value decreased by 35% (green bars), indicating PC and Ca^2+^ crosstalk.

### 3.6. Hypothesis on mPTP Compartmentation

The idea on the important role of variability of mitochondrial responses in the course of ATP-dependent hypercontracture was suggested in [9,11]. This hypothesis implies that preservation of high energy state in one compartment of the cell might balance ATP splitting in a damaged part of cell, where mPTP opening in a fraction of Ca^2+^ loaded mitochondria is needed to run ATP-dependent shortening resulting in round cell shape. The results presented in Figure 6 may support this hypothesis. Here, PC induced temporal hyperpolarization followed by abrupt partial depolarization of ΔΨm (Figure 6A, top cell, red trace), Ca^2+^ overload and hypercontracture in one cell (red hatched box), while in the second cell slow depolarization of ΔΨm over this time period was not accompanied by Ca^2+^ overload (Figure 6, green trace). Remarkably, this partial depolarization was registered in, apparently, an undamaged compartment of hypercontracted oval shape cell outlining the heterogeneity of responses to PC within the same cell.

### 3.7. PC-Induced Ca^2+^ Oscillations

Cytosolic Ca^2+^ oscillations, that occur at the reperfusion phase of I/R, are considered as a component of cell death pathway promoting mPTP and underlying Ca^2+^ overload, hypercontracture, and cell death [11,51]. Inhibition of CaMKII with KN93 was shown to protect CM in I/R-induced spontaneous contractile activity and cell death [51]. In contrast to reperfusion-induced SR-driven Ca^2+^ oscillations, PC-induced high amplitude cytosolic Ca^2+^ oscillations may arise in viable oval shape cells that were resistant to toxic effect of PC excess (Figure 2C). Moreover, the suppression of PLA2 and PLC activities protected against PC-induced Ca^2+^ overload and cell death by increasing the number of oval shaped oscillating cells (Figure 7D–F). At the moment, we cannot answer whether the mechanisms underlying both kinds of Ca^2+^ oscillations are the same or not. This issue requires further investigations.

### 3.8. Implication of Phospholipases in Ca^2+^ Overload and Cell Death

It is well established that the inhibition of phospholipases and CaMKII protect against mPTP opening, necrosis, and apoptosis in I/R injury [46,47,48,49,50,51,52]. In our experiments, combined inhibition of PLA2 and CaMKII provided 60% cell viability in PC-evoked Ca^2+^ overload and cell death (Figure 7A, blue trace). Simultaneous inhibition of PLA2 and PLC ensured 90% cell survival, indicating better protection against PC excess (Figure 7B, blue trace). Activation of Ca^2+^-dependent phospholipases in Ca^2+^-enriched microdomains may represent an early arising event in PC-driven Ca^2+^ overload, hypercontracture, and cell death. Collectively, these results suggest that simultaneous inhibition of cPLA2α and PLCβ,γ could be an effective strategy for protection against PC-mediated toxic effects in cardiomyocytes.

However, the question is whether this kind of “viability” may ensure further restoration of rod shape of precontracted oval shape cells. Besides, what are the mechanisms preventing mitochondrial Ca^2+^ overload and mPTP opening and activation of calpains in contracted cells? Here, we may suggest that both arachidonic acid and IP_3_, which are the products of both phospholipases, might be implicated in the disturbances of Ca^2+^ homeostasis at LCAC excess. Arachidonic acid, and its CoA esters, may reinforce all toxic effects of PC being third player in PC-calcium crosstalk, or acting through its downstream mediators. The inhibition of phospholipases decreased the slope and peak [Ca^2+^]_i_, and, importantly, raised the values of lag-period indicating involvement of both arachidonic acid and IP_3_ in Ca^2+^ overload (Figure 7A,B). To add the complexity, preliminarily, we may suppose the implication of NO/cGMP/kinase G signaling system in registered rhythmic processes and defense mechanisms. However, this issue requires further investigations.

### 3.9. On the Involvement of Ca^2+^-Independent PLA2 (iPLA2) in the Effects of PC

Recently it was demonstrated that activation of mitochondrial Ca^2+^-independent phospholipase A2γ (iPLA2γ) may promote mPTP opening mediating toxic effect via downstream metabolites of arachidonic acid. It was shown that bromenollactone (BEL), the inhibitor of iPLA2, protected mPTP opening induced by Ca^2+^ excess [49]. Taking into account direct and indirect toxic effects of arachidonic acid, we might suppose involvement of iPLA2 in CM death induced by PC or MC. However, our previous results indicate that BEL cannot prevent Ca^2+^ overload and CM death induced by PC, indicating that iPLA2γ is not implicated in the toxic effects of PC observed on isolated CM [26].

## 4. Materials and Methods

All animal procedures were fulfilled in accordance with the EU directive 86/609/EEC and approved by the Ethics Committee at the Institute of Theoretical and Experimental Biophysics, RAS, Russia (Protocol №1, 17.02.20). Male (six-to-eight-week old) Wistar rats were kept under the same conditions in air-conditioned and ventilated rooms at 20–22 °C with a 12 h/12 h light-dark cycle. All experiments were performed at 28 °C.

### 4.1. Mitochondrial Experiments

For mitochondrial assay, mitochondria were isolated without proteolytic digestion from two rat hearts, according to the procedure described earlier [63] using standard techniques of differential centrifugation at 0 °C. Briefly, hearts were perfused with 0.225 M sucrose. The ventricles, chopped and passed through a tissue press, were suspended in an ice-cold solution containing 0.225 M mannitol, 0.075 M sucrose, 5 mM Tris, 0.5 mM EDTA, 0.2 mM ATP, 1% bovine serum albumin fraction V, and pH 7.4 in proportion 8 mL/g tissue. Homogenization was carried out by hand in a glass homogenizer for 30–40 s. The homogenate was centrifuged at 600× *g* for 8 min, and the supernatant was then centrifuged at 13,000× *g* for 10 min. The pellets were finally resuspended in a small glass homogenizer in a medium comprising 225 M mannitol, 0.075 M sucrose, and 2mM TRIS in proportion 0.1 mL/g tissue to yield a suspension containing 35–40 mg protein/mL. The mitochondria incubation medium included: 300 mM sucrose, 50 mM KCl, 3 mM KH2PO4, 10 mM Tris-HCl (pH 7.2), 0.75 mM MgCl2, 0.5 mM EDTA, 10 U/mL hexokinase, 10 mM glucose, and 1% of BSA. The content of mitochondrial proteins was determined by the Lowry method with bovine serum albumin as a standard.

### 4.2. Oxygen Consumption

Oxygen consumption in a mitochondrial suspension was measured by the polarographic method using a Clark-type oxygen electrode in a closed chamber of 2 mL containing 0.3–0.6 mg mitochondrial protein in ml, under continuous stirring.

### 4.3. Mitochondrial Potential

Mitochondrial potential (ΔΨ_m_) changes were evaluated by measuring rhodamine-123 (Rh-123; 1 μM) fluorescence quenching, or by determining the redistribution of lipophilic cation tetraphenylphosphonium (TPP+) between incubation medium and mitochondria. The concentration of TPP+ [TPP+] in the mitochondrial incubation medium was recorded by a TPP+ selective electrode. Changes in calcium ion concentration in the incubation medium were recorded by a Ca^2+^ selective electrode (Nico, Moscow, Russia).

### 4.4. NADH Autofluorescence

The mitochondrial redox state was determined by measuring NAD(P)-linked fluorescence. NAD autofluorescence was excited at 340 nm with HBO lamp, and fluorescence was collected through in 440±10 nm. The NAD(P)H signal increased to a minimum at pO_2_ = 0 and decreased by additional FCCP (1–2 μM). Each signal was calibrated as 0% (pO_2_ = 0) and as 100% (FCCP).

### 4.5. The Opening of the mPTP

The opening of the mPTP was registered as a loss of calcium buffering capacity (a steep rise in calcium in the incubation medium) and/or dissipation of mitochondrial potential ΔΨ_m_. Critical concentration of L-palmitoylcarnitine (PC*), required for the dissipation of ΔΨ_m_, and calcium retention capacity (CRC) were determined as was described earlier [16]. Briefly, the pulses of L-palmitoylcarnitine (10 μM) or CaCl_2_ (50 μM) were added at 60 s intervals. The values of PC* and CRC were determined as total concentrations of added L- palmitoylcarnitine and Ca^2+^ required for mPTP opening. Simultaneous registration of ΔΨ_m_ (with TPP+) and Ca^2+^ was carried out in an open chamber of 1 mL containing 0.3–0.4 mg mitochondrial protein, under continuous stirring.

L-glutamate (10 mM) and pyruvate (0.5–1 mM) or L-malate (1 mM) and a-ketoglutarate (5 mM) were used as substrates to keep the turnover of the tricarboxylic acid cycle. Subsequent addition in the incubation media of 0.75 mM ADP provided a high steady state respiration rate (VO_2_ss), which was close to State 3 respiration rate (80–90% of VO_2max_). L-palmitoylcarnitine (PC). L-myristoylcarnitine (MC), succinate (Succ), pyruvate (Pyr), or beta-hydroxybutytyrate (b-OH) were added after ADP as indicated at subsequent figures.

### 4.6. Epifluorescence and Confocal Microscopy

Cardiomyocytes were placed on 25 mm round coverslips in HBSS, containing 20 mM HEPES, 1.2 mM inorganic phosphate, 1 mM L-Arginine, pH = 7.3 and left for 20–30 min to attach to the glass. Cells were loaded for 40–50 min with 5 µM of AM-ester of calcium-sensitive dyes Fluo-4 or Fura-2 in order to measure [Ca^2+^]_i_, Mag-Fluo-4 to measure [Ca^2+^]_SR_, or Rhod-2 in the case of mitochondrial calcium measurement. Next, staining solution was replaced by dye-free HSBB to 10 min and then cells were washed twice with HBSS and used in experiment. The experiments were carried out using Cell observer imaging system based on Axiovert 200M inverted microscope equipped with AxioCam HSm camera and filtersets no. 44 (Fluo-4), no. 21HE (Fura-2) and 10x/0.3 Plan-Neofluar objective (Carl Zeiss, Germany).

In the case of ΔΨ_m_ measurements the dequench protocol was utilized: cells were loaded with 3 µM TMRM, washed twice and used in experiments. In this case, the increase in TMRM fluorescence measured using filterset no. 45 (Carl Zeiss, Germany) corresponds to mitochondrial depolarization and vice versa.

The XT-images of cardiomyocytes loaded with Fluo-4 were obtained using Leica TCS SP5 confocal microscope (Leica Microsystems, Germany). The Plan Apochromat ×40 objective was used. The scanning was carried out along the long axis of cardiomyocyte with resolution of 300 nm per pixel, at the speed 400 lines per second. Fluo-4 fluorescence was excited with 488 nm line of Argon laser and fluorescence signal was collected in the range of 500–550 nm.

Rhod-2 and Sodium Green Fluorescence was measured using Leica TCS SP5 confocal microscope with the excitation with 543 and 488, respectively. Emission was collected at 560–650 nm for Rhod-2 and 500–530 in the case of SodiumGreen.

Image analysis was performed using ImageJ. After preliminary subtraction of background signal, the ratio F/F_0_ (in some cases ΔF/F_0_) was calculated in the case of single-wavelength dyes and the 340/380 ratio was calculated in the case of Fura-2.

### 4.7. Electrophysiological Experiments

Briefly, currents through the membranes of isolated cardiomyocytes were measured by the whole-cell patch-clamp method in the perforated patch-clamp configuration as described previously [64]. Cardiomyocytes were isolated by proteolytic dispersion of retrograde perfused hearts as described previously [64]. Electrical access to the cardiomyocyte was achieved by perforation of the cell membrane with amfoteritinom B (200–250 μg/mL) added into the glass electrodes with a resistance of 3–5 MOhm, and filled with a solution of the following composition (in mM): CsCl, 130; MgSO_4_, 5; HEPES, 10; pH 7.25. The extracellular bath solution contained (in mM): NaCl, 80; CaCl_2_, 2; MgSO_4_, 5; KH_2_PO_4_, 1.2; CsCl, 10; tetraethylammonium chloride (TEA-Cl), 20; glucose, 20; L-arginine, 1.0; HEPES, 10; pH 7.25. Currents were measured with an Axopatch 200B amplifier (Molecular Devices, United States). The stimulation protocol, calculation of the cell parameters, and the data digitization were carried out with the BioQuest software [64] of the L-154 digital-to-analog/analog-to-digital converter (L-Card, Moscow). Only the cells, in which perforation provided an access resistance of less than 30 MOhm, were used in the experiments.

All reagents were purchased from Tocris (UK) and Sigma (USA).

### 4.8. Statistical Analysis

Statistical analysis of the experimental data was carried out by applying SigmaPlot 11. Student’s *t*-test. *p* < 0.05 was taken as the level of significance. The data are presented by columns as mean ± S.E.M of four to six independent experiments. Compared pairs of values are marked by horizontal lines placed above the columns. Symbol * placed above the lines indicate that *p* < 0.05.

## 5. Conclusions

Here, we demonstrate that long-chain acylcarnitines PC and MC in dose-dependent manner evoke Ca^2+^-sparks and oscillations, long-living Ca^2+^ enriched microdomains, and, finally, Ca^2+^ overload leading to hypercontracture and cardiomyocyte death.

At low concentrations 5 to 10 µM, PC moderately stimulates SR Ca^2+^ release, TCA cycle, and OXPHOS, providing fine Ca^2+^ handling by mitochondria based on SR-mitochondria interplay.

Yet, at high concentrations above 20 µM, PC, like a two-edged sword, further accelerates Ca^2+^ release from SR but limits mitochondrial Ca^2+^ buffering capacity (CRC) by inhibiting TCA cycle (and thus OXPHOS).

Finally, PC induces mitochondrial energetics collapse, and, in combination with Ca^2+^, promotes mPTP opening and Ca^2+^ overload, with minimal impact of external Ca^2+^ influx pathways.

Beside LCAC and calcium, arachidonic acid, or its downstream mediators, and IP3 may also be involved in complex control of Ca2+ homeostasis, reinforcing the effects of LCAC and calcium and collectively resulting in Ca2+ overload and cardiomyocytes death.

Inhibition of Ca^2+^-dependent phospholipases cPLA2α and PLCβ,γ prevents CM death.

Finally, we may suggest that the suppression of both phospholipases may be an effective strategy for the protection against PC-mediated toxicity.

## Figures and Tables

**Figure 1 ijms-21-07461-f001:**
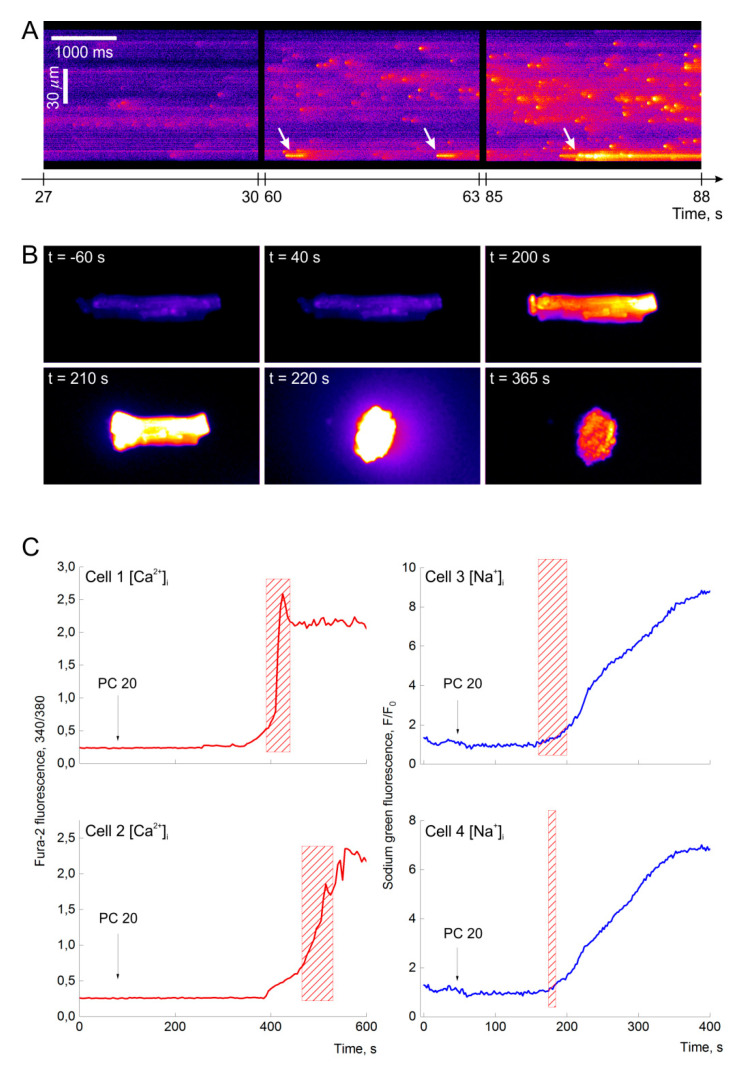
Concentration-dependent effects of PC and MC: Ca^2+^-sparks and Ca^2+^-enriched microdomains, Ca^2+^ overload and CM death. Steep rise in [Ca^2+^]_i_ precedes [Na^+^]_i_ increase. (**A**). At low concentration of 5 µM, myristoyl-L-carnitine (MC) induces Ca^2+^-sparks, short and long time living steady state regimes with high [Ca^2+^]_i_ in isolated microdomains marked by arrows. Confocal images of CM stained with Fluo-4 in XT mode, 27 s (left) and 88 s (right) after the addition of 5 μM MC. Horizontal axis—time; vertical axis—the long axis of the cardiomyocyte. (**B**). At high concentration palmitoyl-L-carnitine (PC, 20 µM) evokes after time delay of approx. 200 s: initial rise of Ca^2+^ in limited compartment of the cell (bright spot at right side of cell, *t* = 200 s), Ca^2+^ overload (bright rod shape cell, *t* = 210 s), hypercontracture and rupture of plasma membrane characterized by fluorescent dye loss (bright oval cell, *t* = 220 s), and cell death (red oval cell, *t* = 365). In 700 and 710 s images the fluorescent probe is seen to escape from the cell into the extracellular solution, leading to decrease in signal intensity. (**C**). Steep rise in [Ca^2+^]_i_ precedes hypercontracture development, while the increment of [Na^+^]_i_ follows it. Representative traces of the changes in cytosolic [Ca^2+^]_i_ (Fura-2, 340/380) and [Na^+^]_i_ (Sodium green, F/F_0_) concentrations throughout the time in 4 cells after the application of 20 µM PC. Red hatched boxes indicate time periods of cell shortening. At the panel, PC 20 means 20 µM PC. Displayed images and graphs are representative of 3 independent experiments.

**Figure 2 ijms-21-07461-f002:**
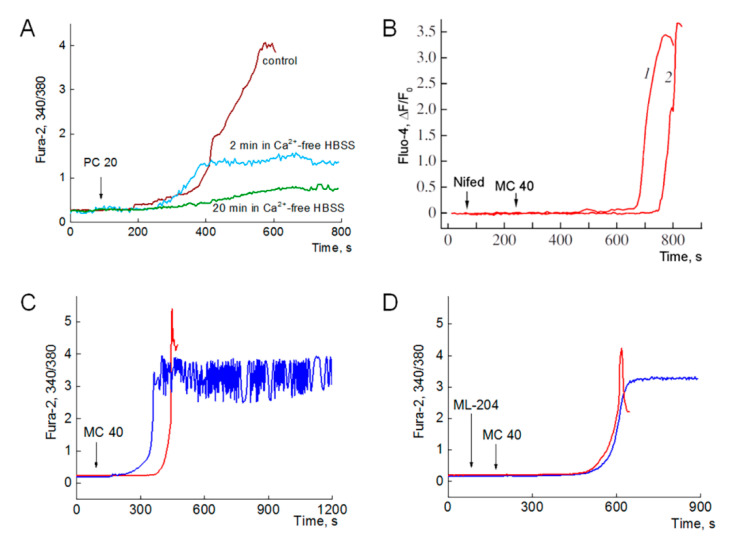
Ca^2+^-overload does not depend on external Ca^2+^ influx. Effects of Ca^2+^-free medium and blockers of LCC and TRPC. (**A**). Effect of extracellular Ca^2+^ on the increment of [Ca^2+^]_i_ (registered as Fura-2 340/380 ratio) induced by 20 µM PC. Here, representative control and typical Ca^2+^ traces are depicted. Cardiomyocytes were exposed to Ca^2+^-free medium for 2 min (red trace) and 20 min (green trace), while brown trace depicts control experiment with 1.3 mM Ca^2+^ in extracellular medium. Hypercontracture and cell death were registered for control cells (*n* = 6) and cells exposed to Ca^2+^-free medium (HBSS) for 2 min (*n* = 8). Cells exposed to Ca^2+^-free medium for 20 min were viable after PC load (*n* = 6). (**B**). L-type calcium channels do not significantly contribute to Ca^2+^ overload. Average values of [Ca^2+^]_i_ (as Fluo-4 ΔF/F_0_) are depicted in pane B for all cells (*n* = 6 and 8). Preincubation of cardiomyocytes with 1 µM nifedipine, a blocker of LCC, did not significantly affected the values of such parameters as lag period, slope, and peak [Ca^2+^]_i_ characterizing Ca^2+^ overload compared to control (traces 1 vs. 2). Ca^2+^ overload, hypercontracture and CM death were induced by 40 µM MC. This panel was adapted from [25] with permission. Viability score (VS), that is the ratio of viable to total cells in the plate, is indicated by corresponding values in panels (**C**,**D**). The effect of TRPC blocker ML-204 (10 µM) on Ca^2+^ overload and Ca^2+^ oscillations evoked by 40 µM MC. Here, representative control (**C**) and typical Ca^2+^ traces registered in presence of ML-204 (**D**) are depicted. In the control 40 µM MC evoked Ca^2+^ overload, hypercontracture and death within 300–350 s in 7 of 9 cells (representative red trace). In 2 cells MC triggered fast Ca^2+^ oscillations lasting over 30 min (dark blue trace).Thus, for the control experiment, VS = 2/9 (*n* = 9). Preincubation of CM with 10 µM ML-204, a potent TRPC blocker, did not have any significant effect on the values of lag period, slope and amplitude of Ca^2+^ rise in 5 dying cells (compared to the parameters of control cells in panel **C**). However, blockade of TRPC abrogated Ca^2+^ oscillations in the rest of the 3 cells that remain viable within 30 min (dark blue representative trace. Thus, for this experiment VS = 3/8 (*n* = 8). Graphs displayed at the panels are representative of 3 independent experiments.

**Figure 3 ijms-21-07461-f003:**
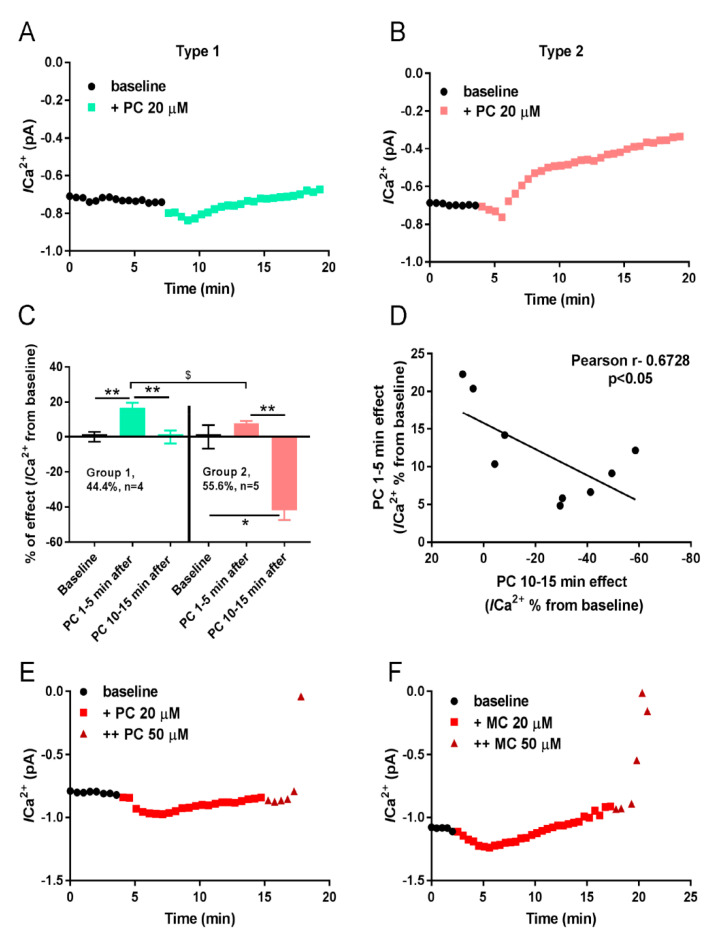
PC exerts multiple effects on peak Ica^2+^ current in ventricular cardiomyocytes. (**A**,**B**). Representative time-courses of peak Ica^2+^ current showing that cardiomyocytes can be separated into two types based on their response to PC. Type 1 (**A**) where PC exerts transient potentiating effect on Ica^2+^, and type 2 (**B**) where PC exerts bidirectional effect: first activating and then inhibiting Ica^2+^. (**C**). Graph bars showing summary of statistical analysis for PC action on Ica^2+^ in two groups of cardiomyocytes. (**D**). Correlation of the level of initial activation of Ica^2+^ with the level of subsequent inhibition in response to PC. ** *p* < 0.05 with ANOVA post hoc Bonferroni test. *P* < 0.01 with Student’s t test, r—Pearson correlation coefficient. (**E**,**F**). Application of 50 µM PC or MC resulted in Ica^2+^ “rundown” accompanied by hypercontracture and CM death (brown triangles). Displayed graphs are representative of 3 independent experiments.

**Figure 4 ijms-21-07461-f004:**
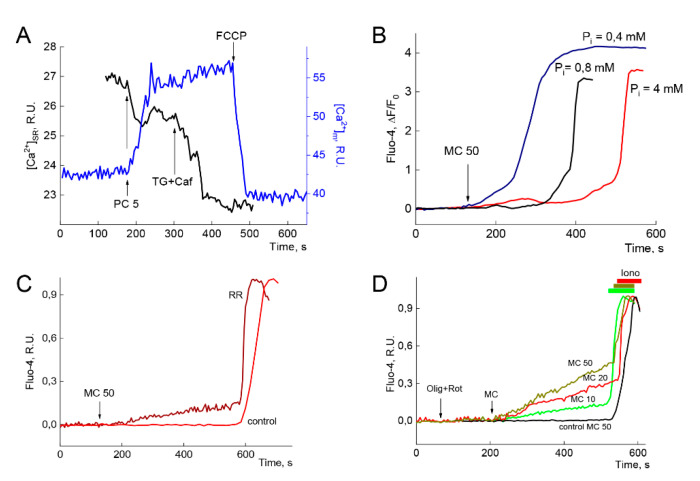
SR-mitochondria interplay. Panel (**A**) depicts redistribution of calcium from SR to mitochondria evoked by PC. At low concentrations, PC (5 µM) induced decrease in SR Ca^2+^ content ([Ca^2+^]_SR_, black line) associated with reciprocal rise in mitochondrial [Ca^2+^]_m_ (blue line) which was measured in another experiment. Two representative Ca^2+^ traces are depicted. *n* = 7, 8. CM were loaded for 30 min with MagFluo-4 or Rhod-2 to monitor the alterations in [Ca^2+^]_SR_ and [Ca^2+^]_m_, respectively. Thapsigargin (100 nM) and caffeine (5 µM) were added to prevent SR Ca^2+^ cycling, which resulted in further in [Ca^2+^]_SR_ lowering indicating correct registration of [Ca^2+^]_SR_. Added uncoupler of OXPHOS FCCP (1 µM) induced rapid loss of mitochondrial [Ca^2+^]_m_ reflecting the dissipation of ΔΨ_m_. Traces presented at the panel are representative of 3 independent experiments. Panel (**B**). Effect of inorganic phosphate (P_i_) on Ca^2+^ buffering in mitochondria and Ca^2+^ overload induced by 50 µM MC. Here, tenfold increase in P_i_ in medium (from 0.4 to 4 mM) resulted in about threefold increment of the lag period, preceding steep rise of [Ca^2+^]_m_ in cytoplasm, indicating modulation of mitochondrial CRC with P_i_. Average values of [Ca^2+^]_i_ (as Fluo-4 ΔF/F_0_) are depicted at pane B for all cells (*n* = 6, 7, 7). This panel was adapted with permission from [25]. Panel (**C**). Effect of ruthenium red (RR, 10 µM), an inhibitor of Ca^2+^ uptake by mitochondria, on Ca^2+^ overload induced by 50 µM MC. This graph shows that moderate inhibition of Ca^2+^ uptake via MCU abrogated fine Ca^2+^ handling in the cytosol. Cytosolic Ca^2+^ slowly increased till the moment of Ca^2+^ overload, while the slope and maximum [Ca^2+^]_i_ were not modified by RR (brown vs. red traces). Average values of [Ca^2+^]_i_ (as Fluo-4 ΔF/F_0_) are depicted in pane **C** for all cells (*n* = 6, 8). This panel was adapted with permission from [25]. Panel (**D**). Inhibition of OXPHOS with oligomycin (1 µM) and rotenone (1 µM) prevented Ca^2+^ buffering in mitochondria resulting in concentration-dependent Ca^2+^ accumulation in cytosol. Concentrations of added MC are indicated on the traces (in µM). Control experiment (50 µM MC, black line) was performed in the medium without oligomycin and rotenone. Application of ionomycin (1 µM, coloured boxes, Iono) was used to normalize Ca^2+^ responses in the experiments with oligomycin and rotenone (color lines). Here, representative Ca^2+^ traces are depicted. *n* = 6–10. Traces presented at each panel of Figure 4 depict one of 3 independent experiments.

**Figure 5 ijms-21-07461-f005:**
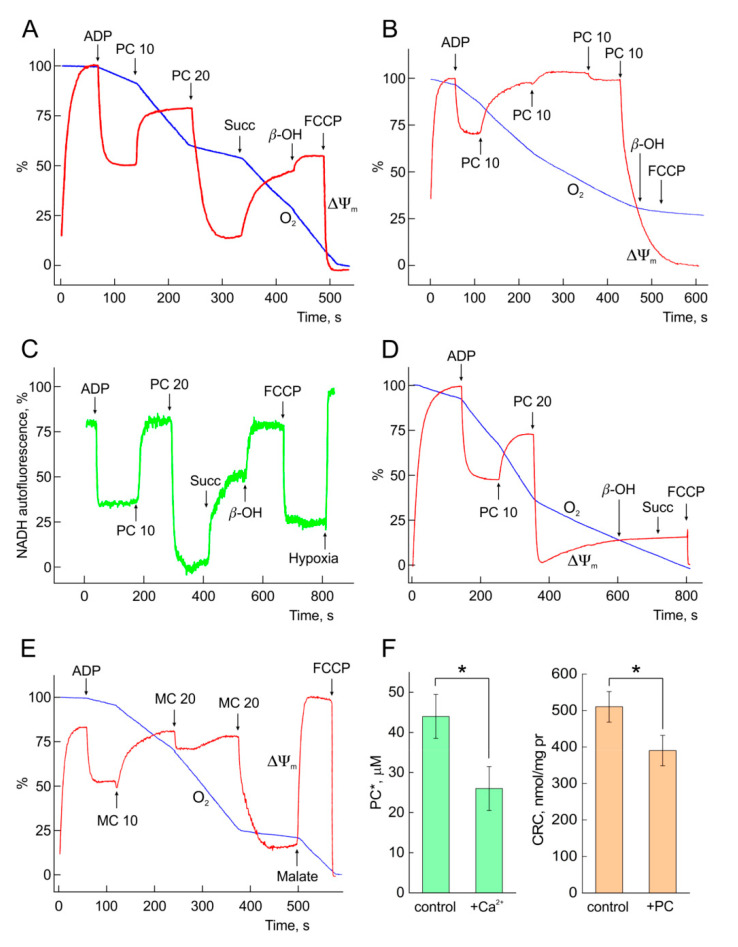
Concentration-dependent effects of PC or MC on OXPHOS in isolated mitochondria. Activation, reversible and irreversible inhibition of OXPHOS and TCA cycle by PC and MC in the experiments on isolated rat heart mitochondria. In panels (**A**,**B**,**D**,**E**) are presented mitochondrial oxygen consumption (O_2_ traces, blue) and potential (ΔΨ_m_, red). Panel (**C**) depicts the alterations of mitochondrial NAD(P)H fit to the experiment of panel (**A**). Oxygen consumption in a mitochondrial suspension was measured by Clark-type oxygen electrode in a closed chamber of 2 mL wth 0.3–0.6 mg mitochondrial protein in ml, under continuous stirring. Mitochondrial potential (ΔΨ_m_) changes were evaluated by measuring rhodamine-123 (Rh-123) fluorescence, except for panel F where TPP+ was recorded with TPP+ selective electrode in open chamber. ADP (0.5 mM) was added at the beginning of each experiment to preset steady state mitochondrial respiration (in presence of hexokinase and glucose in incubation medium). Malate (1 mM) plus α-ketoglutarate (5 mM) were used as substrates in the experiments in panels (**A**,**C**). Glutamate (10 mM) plus pyruvate (0.5 mM) were used as substrates in the experiments in panels (**B**,**D**,**E**,**F**). Values of PC and MC are in µM. Panels (**A**,**C**,**E**) show the examples of reversible inhibition of OXPHOS and TCA cycle by PC (panels **A**,**C**) and MC (panel **E**). Here, addition of succinate (5 mM), malate (5 mM), and β-oxybutyrate (b-OH, 10 mM) provide reactivation of the respiration (slope of O_2_ trace, red), rise of ΔΨ_m_ (blue) and NADH (green). Meanwhile, panels (**B**,**D**) illustrate irreversible inhibition of OXPHOS by PC excess when addition of succinate and β-oxybutyrate is not effective. Panels A to E display one of 3 independent experiments. Panel (**F**) depicts average for 5 experiments values of both, threshold concentration of PC required for mPTP opening (PC*) and of mitochondrial calcium retention capacity (CRC). This panel describes the effect of 20 µM Ca^2+^ on PC* and, vice versa, of 20 µM PC on CRC. * *p* < 0.05 with ANOVA post hoc Bonferroni test. *p* < 0.01 with Student’s *t*-test.

**Figure 6 ijms-21-07461-f006:**
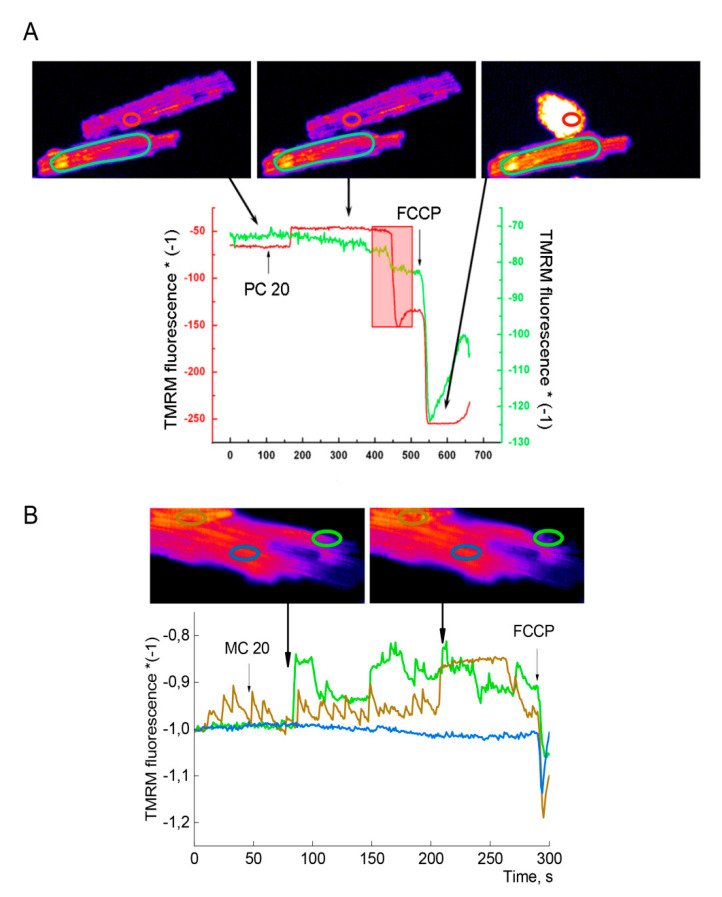
Variability of ΔΨ_m_ responses to PC in isolated CM. Variety of ΔΨ_m_ responses to 20 µM PC in two cells (panel **A**) and to 20 µM MC in nearby compartments of one cell (panel **B**). TMRM fluorescence, as a measure of ΔΨ_m_, is depicted in red and green traces for top and bottom cells in panel A, respectively. In panel (**B**), 3 colored TMRM traces correspond to signals from 3 limited oval areas at one cell. Uncoupler FCCP (1 µM) wad added for dissipation of ΔΨ_m_. Red hatched box corresponds to time period of top cell contraction evoked by PC (**A**).

**Figure 7 ijms-21-07461-f007:**
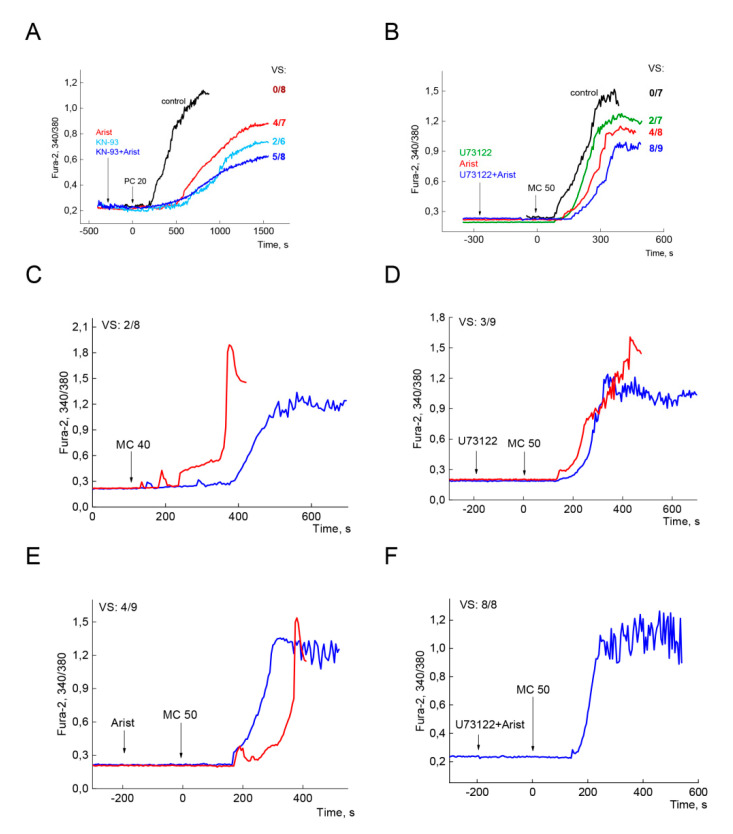
Implication of phospholipases and CaMKII in Ca^2+^ overload and myocytes death. Panel (**A**). Separate and combined effects of aristolochic acid (Arist, 100 µM) and KN93 (20 µM), inhibitors of PLA2 and CaMKII, respectively, on Ca^2+^ overload and CM death/viability at 20 µM PC. Viability score (VS) is the ratio of viable to total cells (*n*) in the plate. Average values are depicted for all dead cells in control (black line, VS = 0/8, *n* = 8) and all viable cells for traces with the inhibitors (colored lines). For Arist, KN93, and Arist + KN93 viability score VS = 4/7, 2/6, and 5/8, respectively. *n* = 7, 6, and 8. Panel (**B**). Separate and combined effects of U73122 (10 µM), inhibitors of PLC, and Arist, on Ca^2+^ overload and CM death/survival at 50 µM MC. Average values are depicted for all dead cells in control (black line, VS = 0/7, *n* = 7) and all viable cells for traces with the inhibitors (colored lines). For U73122, Arist, and U73122 + Arist viability score VS = 2/7, 4/8, and 8/9, respectively. *n* = 7, 8, and 9. All displayed graphs are representative of 3 independent experiments. Panels (**C**–**F**) depict representative traces characterizing dying cells (C, red traces) and viable cells (dark blue traces) in the populations of cells predisposed to Ca^2+^ oscillations and partially resistant to MC or PC excess. Panel (**C**) depicts control experiment with 50 µM MC (VS = 2/8, *n* = 8). Separate and combined effects of U73122 and Arist are presented in panels (**D**–**F**). Here, VS = 3/9, 4/7, and 8/8, respectively. *n* = 9,7, and 8, respectively. Traces presented at the panels depict of 1 of 3 independent experiments.

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
