# Peer review of "Dissecting Cellular Mechanisms of Long-Chain Acylcarnitines-Driven Cardiotoxicity: Disturbance of Calcium Homeostasis, Activation of Ca2+-Dependent Phospholipases, and Mitochondrial Energetics Collapse"

_ijms, 2020, doi:10.3390/ijms21207461_

Round 1

Reviewer 1 Report

This manuscript represents solid and comprehensive study of cadiomyocyte response to long-chain fatty acids, in particular long-chain acylcarnitines, known to be involved in ischemia-reperfusion (I/R) induced myocardial injury. Positive feature of the present study is the implementation of a spectrum of imaging techniques addressing diverse intracellular and mitochondrial functions, such as Ca2+ handling, mitochondrial membrane potential, oxidative phosphorylation, etc.

Minor concerns:

Some critique may be raised against measurements of L-type Ca2+ current through channels incorporated in the lipid bilayer, under condition when the channel proteins unlinked from regulation via changed in the intracellular kinase/phosphatase balance that, in turn, can be subject for changes in energy cellular status. On the other hand, direct effect of long-chain acylcarnitines on L-type Ca2+ channels, although, as the authors demonstrated, has a weak impact, may be of particular interest.

The authors may wish to provide a more clear statement (presumably at the beginning of Discussion) of their main discoveries and interpret their data in line with the introductory premise addressing the mechanisms of cardiac dysfunction under acute ischemia/reperfusion. How uncovered by the authors effects of low and high doses of long-chain acylcarnitines correspond to acute ischemic events? How identified cellular and mitochondrial responses may contribute to a time course of ischemia and particularly during reperfusion? Such discussion would definitely improve an impact of the manuscript.

The text, seems to me, is overloaded with abbreviations. Avoiding, where it possible, usage of multiple abridgements in one sentence would definitely improves the readability.

Author Response

To Reviewer 1

Remark 1

Some critique may be raised against measurements of L-type Ca2+ current through channels incorporated in the lipid bilayer, under condition when the channel proteins unlinked from regulation via changed in the intracellular kinase/phosphatase balance that, in turn, can be subject for changes in energy cellular status. On the other hand, direct effect of long-chain acylcarnitines on L-type Ca2+channels, although, as the authors demonstrated, has a weak impact, may be of particular interest.

Re: We used perforated patch clamp to register L-type Ca2+ current in freshly isolated cardiomyocytes and only refer to the results obtained by another group based on the measurements of L-type Ca2+ current through channels incorporated in the lipid bilayer.

Remark 2

The authors may wish to provide a more clear statement (presumably at the beginning of Discussion) of their main discoveries and interpret their data in line with the introductory premise addressing the mechanisms of cardiac dysfunction under acute ischemia/reperfusion. How uncovered by the authors effects of low and high doses of long-chain acylcarnitines correspond to acute ischemic events?

How identified cellular and mitochondrial responses may contribute to a time course of ischemia and particularly during reperfusion? Such discussion would definitely improve an impact of the manuscript.

Re:

We appreciate this recommendation to improve the formulations of our results. List of  basic statements is incorporated in Conclusions. As for expected effects of low and high doses of long-chain acylcarnitines in I/R, recognizing that this is hot and not studied issue, we briefly addressed it in Discussion.

Remark 3

The text, seems to me, is overloaded with abbreviations. Avoiding, where it possible, usage of multiple abridgements in one sentence would definitely improves the readability.

Re:

Part of abbreviations and, especially, few abridgements used within one sentence, were removed.

Reviewer 2 Report

In this paper, the authors investigated the mechansim by which acute LCAC exposure triggers cardiomyocyte death. They observe that PC (or MC) exposure induces cytosolic Ca2+ rises, likely as a consequence of SR Ca2+ release, in turn sensed by mitochondria which undergo IMM depolarization associated with cell death. Though these observations are not completely novel, overall I think the paper is interesting, because it deepens some aspects that so far have been fragmentarily investigated.

However, I have several concerns that need to be addressed before publication in this journal.

  • The continuous shift between the use of PC and MC is confusing. Also, what does “PC (MC)” mean (see for example lines 124)? Have been PC and MC used together or alone? If alone, only half of the experiments are shown. Please add these experiments or remove PC/MC accordingly.
  • In different points, experiments from previous publications are cited and used as control to move forward with the paper (see for example line 121-122, or fig 4B which is adapted with permission from ref 25). While this is acceptable in some cases, I think that at least key experiments should be repeated.
  • Lines 136-137: I think that the delay in [Na+] rise does not necessarily indicate mPTP opening. Note that Ca2+ elevations may induce cell death also by additional mechanisms (such as calpain activation) that should be considered.
  • In general, the number of cells used in the different experiments is low (<10). I see that increasing these numbers might be difficult, especially considering that different assays have been performed, but in some cases the authors reach conclusions that are not convincingly sustained by these numbers. As an example, see lines 198-206. The fact that 2 out of 9 control cells display Ca2+ oscillations, whereas upon TRPC inhibition no oscillations are observed in 3 out of 8 cells, is not really impressive. Indeed, with so small numbers (e.g. 2 events), it is not easy to understand whether this occurs by chance or it is significant. Specific statistic tests should be used to address this point. If necessary, please increase n, or remove accordingly if the tests are not significant.
  • In fig. 4a, the drop in SR Ca2+ and the rise in mitochondrial Ca2+ are almost immediate upon PC (5uM) addition. Surprisingly, in the cytosol, a lag of approximately 5 min exists before observing a minimal rise of [Ca2+] (see for example fig 1c, where an even higher PC concentration has been used). What is the explanation? I don’t think that the close SR-mitochondria apposition can be invoked in this case, because the fluorescent indicators used to monitor cytosolic Ca2+ (FURA and Fluo) are sensitive enough to reveal also small Ca2+ microdomains (see Fig 1A). A tiny increase in [Ca2+]i should be observed.
  • Again, the continuous change between the use of PC and MC, as well as the change in their concentrations, does not allow to properly compare the effects in the different cell compartments and among different experiments (see for example Fig. 4a and 4 B). Similarly, why in some experiments FURA has been used, in others Fluo3 and in others Fluo4?   
  • 5: please add the n for all panels.
  • Fig 7 C to F: I have some problems to interpret the significance of the oscillating cells. Are there significant differences between the conditions represented in each panels? For example, in panel D, is there a statistically significant difference between the condition represented by the dark blue trace and that of the cyan trace? Similarly in the remaining panels.
  • Paragraph 2.4.1: Please refer to fig. 7 and not 6.
  • Data on phospholipases are interesting but possibly undermine the role of Ca2+ for the death process. I see that PLC is closely linked to Ca2+ signaling, but the fact that the combined phospholipase inhibition protects cells from death, while maintaining sustained Ca2+ rises (fig. 7), questions the importance of Ca2+. I suggest to perform a control by incubating cells with BAPTA-AM to buffer intracellular Ca2+ and check the effect on cell death.
  • Why in fig. 4D the rise in cytosolic Ca2+ is so slow compared with fig. 4B? Note that in Fig 4D mitochondrial ATP synthesis is inhibited. Therefore, the lower cellular ATP levels should aggravate Ca2+ overload, because different ATP-dependent pumps help cells to maintain Ca2+ homeostasis, by extruding Ca2+ or taking it up in the SR. In the methods, it is reported that the experiments have been performed in HBBS, i.e., with a sub-millimolar [Pi] (such as in the blue or black trace in fig. 4B, i.e. those in which Ca2+ rise occurs early).  
  • Lines 532-534: I don’t see the experiments with BEL. Please remove this statement if they are not available.

Author Response

To Reviewer 2

Remark 1

The continuous shift between the use of PC and MC is confusing. Also, what does “PC (MC)” mean (see for example lines 124)? Have been PC and MC used together or alone? If alone, only half of the experiments are shown. Please add these experiments or remove PC/MC accordingly.

Re:

All PC/MC and PC (MC) were removed in the text.

Remark 2

In different points, experiments from previous publications are cited and used as control to move forward with the paper (see for example line 121-122, or fig 4B which is adapted with permission from ref 25). While this is acceptable in some cases, I think that at least key experiments should be repeated.

Re: In august we performed some experiments to be sure that key earlier results are reproducible. At a moment we cannot run the experiments which may occupy several months. As for the effect of inorganic phosphate Pi (Fig. 4D), modulation of mitochondrial calcium buffering capacity with Pi may be observed both in the cells and isolated mitochondria. In 2003 bimodal effect of Pi on CRC was shown for the first time on isolated mitochondria in [40].

Remark 3

Lines 136-137: I think that the delay in [Na+] rise does not necessarily indicate mPTP opening. Note that Ca2+ elevations may induce cell death also by additional mechanisms (such as calpain activation) that should be considered.

Re:

We appreciate this important remark. Categorical statement about pore opening (lines 142-144) is eliminated and replaced by the possible involvement of this mechanism. As for calpains activation, we attempted to discuss briefly this mechanism in Discussion in relation to phospholipases and Ca2+ oscillations when cytosolic Ca2+is high.

Remark 4

In general, the number of cells used in the different experiments is low (<10). I see that increasing these numbers might be difficult, especially considering that different assays have been performed, but in some cases the authors reach conclusions that are not convincingly sustained by these numbers. As an example, see lines 198-206. The fact that 2 out of 9 control cells display Ca2+ oscillations, whereas upon TRPC inhibition no oscillations are observed in 3 out of 8 cells, is not really impressive. Indeed, with so small numbers (e.g. 2 events), it is not easy to understand whether this occurs by chance or it is significant. Specific statistic tests should be used to address this point. If necessary, please increase n, or remove accordingly if the tests are not significant.

Re: Having some experience in the studies of rhythmic processes, nevertheless, previously we ignored Ca2+ oscillations evoked by PC suggesting that this is trivial feature of positive feedback systems. Now, evaluating possible implication of Ca2+ oscillations in I/R (ref. [11, 51]), arrythmia, etc., we suppose that positive feedbacks with PLC and PKG may underlie such kind of rhytmicity (but not “Ca2+-clock” mechanism after Lacatta et al). Here, analyzing previous and recent data, we have found that rare quasi-periodic Ca2+ oscillations may be observed in some cells (in 10 to 15% of all experiments). It may be 2-3 of 7-10 cells in a coverslip, which are resistant to PC toxicity and contract in spite of high average Ca2+ level (Fig. 2C, Figs. 7C-7F). Each of these graphs is representative of 3 independent experiments. But, we cannot predict beforehand in which experiment part of cells will display rhytmicity evoked by PC. As for TRPC 4,5 channels, the panels C and D of Fig.2 simply demonstrate that Ca2+ influx in SR may be important for periodic mode operation of the system. In the models of simplest positive feedback systems the value of influx may preset periodic or steady state operation.

Remark 5

In fig. 4a, the drop in SR Ca2+ and the rise in mitochondrial Ca2+ are almost immediate upon PC (5uM) addition. Surprisingly, in the cytosol, a lag of approximately 5 min exists before observing a minimal rise of [Ca2+] (see for example fig 1c, where an even higher PC concentration has been used). What is the explanation? I don’t think that the close SR-mitochondria apposition can be invoked in this case, because the fluorescent indicators used to monitor cytosolic Ca2+ (FURA and Fluo) are sensitive enough to reveal also small Ca2+ microdomains (see Fig 1A). A tiny increase in [Ca2+]i should be observed.

Re: Please take into consideration that 5uM PC induced only 5-10% alterations in both SR Ca2+ (27 to 25.5 R.U.) and mitochondrial Ca2+ (43 to 54 R.U), while TG+ Caffeine provide further fall of SR Ca2+ sufficient to be monitored with FURA.

Remark 6

Again, the continuous change between the use of PC and MC, as well as the change in their concentrations, does not allow to properly compare the effects in the different cell compartments and among different experiments (see for example Fig. 4a and 4 B). Similarly, why in some experiments FURA has been used, in others Fluo3 and in others Fluo4?

Re:  Fluo3 was erroneously indicated at Fig. 4C. Thanks, appropriate corrections were made. The experiments with FURA and Fluo4 were performed in different time periods and on different microscopes. As for the concentrations of PC and MC used at Fig. 4A and 4 B, Fig. 4A demonstrates that even minimal concentration of 5uM bring about to the alterations in SR and mitochondrial Ca2+. While, Fig. 4 B displays the effect of variation of mitochondrial Ca2+ buffering observed at maximal concentration MC (50 uM).

Remark 7

5: please add the n for all panels.

Re: n was added for all panels.

Remark 8

Fig 7 C to F: I have some problems to interpret the significance of the oscillating cells. Are there significant differences between the conditions represented in each panels? For example, in panel D, is there a statistically significant difference between the condition represented by the dark blue trace and that of the cyan trace? Similarly in the remaining panels.

Re: At the panels C through D blue, dark blue and cyan traces characterize viable oscillating cell. Now, to avoid misinterpretation, we indicate here only two representative traces for dying (red) and oscillating (blue) cells on each panel.

Remark 8

Paragraph 2.4.1: Please refer to fig. 7 and not 6.

Re:Thanks, now we refer to Fig. 7 at paragraph 2.4.1

Remark 9

Data on phospholipases are interesting but possibly undermine the role of Ca2+ for the death process. I see that PLC is closely linked to Ca2+ signaling, but the fact that the combined phospholipase inhibition protects cells from death, while maintaining sustained Ca2+ rises (fig. 7), questions the importance of Ca2+. I suggest to perform a control by incubating cells with BAPTA-AM to buffer intracellular Ca2+ and check the effect on cell death.

Re: We appreciate this remark and proposal to buffer intracellular Ca2+. We are planning such kind of experiments and expect that arachidonic acid (as extra load, in addition to PC) and, may be, its downstream metabolites are involved too.

Remark 10

Why in fig. 4D the rise in cytosolic Ca2+ is so slow compared with fig. 4B? Note that in Fig 4D mitochondrial ATP synthesis is inhibited. Therefore, the lower cellular ATP levels should aggravate Ca2+ overload, because different ATP-dependent pumps help cells to maintain Ca2+ homeostasis, by extruding Ca2+ or taking it up in the SR. In the methods, it is reported that the experiments have been performed in HBBS, i.e., with a sub-millimolar [Pi] (such as in the blue or black trace in fig. 4B, i.e. those in which Ca2+ rise occurs early).

Re: Thank yoy for this remark. Erroneously we did not indicate that our medium included in total 2 mM Pi to avoid weak Ca2+ buffering at non physiological low Pi in HBBS (0.4 mM). After ischemia Pi may rise to 20-25 mM. Besides, we run all our experiments with 1mM L-arginine to minimize run-down and stabilize L-current responses (in patch) and switch on NO/cGMP/PKG- signaling system.

Remark 11

Lines 532-534: I don’t see the experiments with BEL. Please remove this statement if they are not available.

Re: We excluded this paragraph and considered briefly possible impact of BEL (iPLA2γ) in Discussion referring to our previous results.

Reviewer 3 Report

Berezhnov and colleagues have presented interesting experiments to explore the impact of PC and MC on cardiomyocytes regarding calcium handling and mitochondrial metabolism. I specifically enjoyed the interaction between SR calcium handling and mitochondria. Furthermore, the visualization of calcium-enriched microdomains in cardiomyocytes was quite impressive.

  1. Section 2.1.2:
  2. I do not see that a steep rise in [Ca2+]i (400 s) precedes [Na+]i increase (200 s). Ca2+ and Na+ measurements were performed in different cells? The red boxes in the Na+ diagrams also indicate hypercontracture? Because this is seen before a rise in calcium.
  3. What is seen in Figure 1A: MC or PC treatment? Figure legend describes both.

  1. Section 2.2.1 SR experiments (Figure 2): Perhaps the authors could further investigate the SR calcium stores by applying caffeine. The authors may explore the effect of caffeine in combination with PC in calcium free buffer (for 2 min) on mPTP opening. This may support the hypothesis that SR calcium and PC is enough to induce calcium overload.

  1. Section 2.2.2.: The authors described that L-type calcium and transient receptor potential channels do not significantly contribute to Ca2+ I just have some questions regarding the experimental set up: Did the cardiomyocytes contracted spontaneously or were they electrically stimulated? Why did ML-204 prevented calcium oscillations in the viable cells? Did these cells stopped contracting? The authors use ML-204 to investigate the involvement of TRPC. Yet, ML-204 is described to only block TRPC4. Do cardiomyocytes also express other TRPC subtypes? If yes, is ML-204 appropriate for the experiments?

  1. Section: 2.2.3.: Interestingly, the authors observed different effects of PC on L-type calcium current. Do the authors have any explanation? I would be careful to call it biphasic effect since the cells revealed either an activation or an inhibition.

  1. Section: 2.3.2 Sarcoplasmic reticulum (SR) – mitochondria interplay. In Figure 4 A the authors demonstrated that PC evoked the redistribution of calcium from the SR to mitochondria. Just one question regarding the SR-mitochondria interplay: How would the alterations in [Ca2+]SR and [Ca2+]m look using only tapsigargin and caffeine?

Figure 4D: Why did the authors use different Flou dyes? Fluo-3 and Fluo-4? The conclusion for the experiment in 4 D regarding the modulation of RyR channels activity by PC became not clear. Could it also be an inhibition of SERCA? I just would be careful with this conclusion and perhaps only indicate an involvement of SR calcium? Please describe the use and purpose of ionomycine, otherwise the results are confusing.

  1. Section 2.3.3 and 4: Very nice section.

  1. Section 2.3.5: The authors carefully distinguished between distinct areas of the cell and the effect of PC/MC on mitochondrial ΔΨm. Based on the experimental set-up it seems that the effect on TMRM fluorescence is due to distinct changes of intracellular calcium handling (compartmentation) and not due to different types of mitochondria (written in results part 447-448). Perhaps rephrase last sentence in the results?

  1. Section 2.4.1: Wrong Figure number in text: Line 486 Figure 6, but I think the authors mean Figure 7. Line 487: Authors describe black line in Figure 7 A, but there is no black line. Figures in 7 have different sizes. Line 498: brown traces, I think the authors mean red. Please check all color descriptions in text and figure.

I apologize in advance for the following questions, I have only very little experiences with such experiments: Why is the calcium level in the cells constantly high? It never reaches baseline levels but the cells are still viable and contract (independent of drug)? Does this argue for a constant calcium leak induced by PC/MC?

  1. In the abstract the authors suggest that simultaneous inhibition of cPLA2 and PLC could be an effective strategy for protection against PC-mediated toxic effects in cardiomyocytes. Long-chain ACs are lipophilic metabolites and have therefore detergent-like effects. I am wondering whether a further accumulation of PC would result also in cardiomyocyte death due to membrane integration and disruption as reported by others. The L-type calcium channel experiments with high PC/MC concentrations resulted in abrupt calcium rundown and cell death suggesting also membrane disruption. Therefore, avoiding long-chain AC accumulation might be also relevant regarding protection against PC-mediated toxicity (CPT1 inhibitor?)?

  1. The authors divided the discussion in 7 parts. This is only a suggestion, but perhaps a more continuous discussion (with less subparts) would help to understand the connection between the results. Furthermore, there is a lot of repetition of the results.

Minor comments:

  1. Abbreviation CMC for cardiac muscle cells. I would suggest to use only CM (cardiomyocytes) as abbreviation since CMC also stands for critical micelle concentration which is quite relevant in the field of acyl-carnitines.
  2. Line 168: it is written red vs. black line in Figure 2A. There is no black line.
  3. Figure 2 B: Perhaps use different colors (just a suggestion)
  4. Line 201: blue and violet traces in Figure 2 C. There is no violet trace.
  5. Line 235: path-clamp. Should be patch-clamp?
  6. Several diagrams differ in their size.

Author Response

To Reviewer 3

Remark 1

1.Section 2.1.2:

  1. I do not see that a steep rise in [Ca2+]i (400 s) precedes [Na+]i increase (200 s). Ca2+ and Na+ measurements were performed in different cells? The red boxes in the Na+ diagrams also indicate hypercontracture? Because this is seen before a rise in calcium.

Re:  We appreciate this remark. Of course, Ca2+ and Na+ measurements were performed in different cells. The red boxes in both Ca2+ and Na+ diagrams indicate hypercontracture. Now, to avoid confusion, we indicated this not only in the legend but also at the panels: Cells 1 to 4.

  1. What is seen in Figure 1A: MC or PC treatment? Figure legend describes both.

Re: Panel A corresponds to experiment with 5uM MC as indicated in its legend, while panes B and C – with  20 uM  PC.

  1. Section 2.2.1

SR experiments (Figure 2): Perhaps the authors could further investigate the SR calcium stores by applying caffeine. The authors may explore the effect of caffeine in combination with PC in calcium free buffer (for 2 min) on mPTP opening. This may support the hypothesis that SR calcium and PC is enough to induce calcium overload.

Re: We appreciate this proposal.

3.Section 2.2.2.:

The authors described that L-type calcium and transient receptor potential channels do not significantly contribute to Ca2+ I just have some questions regarding the experimental set up: Did the cardiomyocytes contracted spontaneously or were they electrically stimulated? Why did ML-204 prevented calcium oscillations in the viable cells? Did these cells stopped contracting? The authors use ML-204 to investigate the involvement of TRPC. Yet, ML-204 is described to only block TRPC4. Do cardiomyocytes also express other TRPC subtypes? If yes, is ML-204 appropriate for the experiments?

 Re: Thank you, we had to indicate that is ML-204 at 3-10 uM is selective blocker of TRPC4,5 channels involved with TRPC1 in SOCE. The cardiomyocytes also express TRPC1, 3, and 6. Blocker SKF96365 is non selective influencing all these and TRPM channels. So, we used ML-204 to investigate the impact of TRPC4,5. Now, we included appropriate remarks in the text indicating only TRPC4,5 channels. We used in our experiments non contracting spontaneously and not stimulated cardiomyocytes.

Here, PC evoked rare quasi-periodic Ca2+ oscillations that may be observed in some cells (in 10 to 15% of all experiments). It may be 2-3 of 7-10 cells in a coverslip, which are resistant to PC toxicity and contract in spite of high average Ca2+ level (Fig. 2C, Figs. 7C-7F). Ca2+ oscillations may be implicated in I/R (ref. [11, 51]), arrythmia, etc. That is why we register such regimes to analyze in future possible mechanisms. In positive feedback systems, generating such kind of regimes, periodic or steady functioning is preset by the value of influx. Here, inhibition of TRPC4,5 (Ca2+ influx in SR) prevented calcium oscillations in the cell and switched Ca2+ signaling machinery into steady state regime.

4.Section: 2.2.3.

Interestingly, the authors observed different effects of PC on L-type calcium current. Do the authors have any explanation? I would be careful to call it biphasic effect since the cells revealed either an activation or an inhibition.

Re: We agree that biphasic effect may be referred to the cells of second type. Appropriate corrections are introduced in the text. At a moment we have no idea on the mechanisms underlying this difference.

5.Section: 2.3.2

1.Sarcoplasmic reticulum (SR) – mitochondria interplay. In Figure 4 A the authors demonstrated that PC evoked the redistribution of calcium from the SR to mitochondria. Just one question regarding the SR-mitochondria interplay: How would the alterations in [Ca2+]SR and [Ca2+]m look using only tapsigargin and caffeine?

Re: Earlier experiments demonstrate that tapsigargin and caffeine provide [Ca2+]m  accumulation [11] based on slow rise of [Ca2+]I [25].

  1. Figure 4D: Why did the authors use different Flou dyes? Fluo-3 and Fluo-4? The conclusion for the experiment in 4 D regarding the modulation of RyR channels activity by PC became not clear. Could it also be an inhibition of SERCA? I just would be careful with this conclusion and perhaps only indicate an involvement of SR calcium? Please describe the use and purpose of ionomycine, otherwise the results are confusing.

Re: Fluo3 was erroneously indicated at Fig. 4C. Thanks, appropriate corrections were made. We agree that more correct would be the statement on the impact of PC on SR calcium cycling. It is known that avian SERCA may be inhibited by PC above 20 uM (Dumonteil E, Barré H, 1994). Ionomycin was applied to normalize the responses to max values. We introduced appropriate corrections in the text.

6.Section 2.3.3 and 4: Very nice section.

7.Section 2.3.5:

The authors carefully distinguished between distinct areas of the cell and the effect of PC/MC on mitochondrial ΔΨm. Based on the experimental set-up it seems that the effect on TMRM fluorescence is due to distinct changes of intracellular calcium handling (compartmentation) and not due to different types of mitochondria (written in results part 447-448). Perhaps rephrase last sentence in the results?

Re: We agree and did this. 

  1. Section 2.4.1:

 Wrong Figure number in text: Line 486 Figure 6, but I think the authors mean Figure 7. Line 487: Authors describe black line in Figure 7 A, but there is no black line. Figures in 7 have different sizes. Line 498: brown traces, I think the authors mean red. Please check all color descriptions in text and figure.

 Re: We agree, correct number of the figure and right color of the lines are introduced in the text. 

  1. I apologize in advance for the following questions, I have only very little experiences with such experiments: Why is the calcium level in the cells constantly high? It never reaches baseline levels but the cells are still viable and contract (independent of drug)? Does this argue for a constant calcium leak induced by PC/MC?

 Re: Yes, we agree that calcium leak is required for this effect. Hypercontracture and death are observed in most of cells. But, some rare cells are resistant being round shape and viable at rather high [Ca2+]I (fig. 2D). The same is true with protection provided by phospholipases inhibitors (fig. 7). Here, PC, Ca2+, and arachidonic acid interplay , as we suggest in discussion, may take place.

  1. In the abstract the authors suggest that simultaneous inhibition of cPLA2 and PLC could be an effective strategy for protection against PC-mediated toxic effects in cardiomyocytes. Long-chain ACs are lipophilic metabolites and have therefore detergent-like effects. I am wondering whether a further accumulation of PC would result also in cardiomyocyte death due to membrane integration and disruption as reported by others. The L-type calcium channel experiments with high PC/MC concentrations resulted in abrupt calcium rundown and cell death suggesting also membrane disruption. Therefore, avoiding long-chain AC accumulation might be also relevant regarding protection against PC-mediated toxicity (CPT1 inhibitor?)?

 Re: We absolutely agree with your remarks and statements. At 70 -80 uM,  MC displays detergent-like effects. CPT1 inhibitors were recommended about 30 years ago to protect myocardium (mitochondria) of fatty acids toxic effect. Long – time based treatment with meldonium- like preparations looks promising too.

11.The authors divided the discussion in 7 parts. This is only a suggestion, but perhaps a more continuous discussion (with less subparts) would help to understand the connection between the results. Furthermore, there is a lot of repetition of the results.

Re: Thank, as far it was possible we tried to eliminate these shortcomings.

  1. Minor comments:
  1. Abbreviation CMC for cardiac muscle cells. I would suggest to use only CM (cardiomyocytes) as abbreviation since CMC also stands for critical micelle concentration which is quite relevant in the field of acyl-carnitines.
  2. Line 168: it is written red vs. black line in Figure 2A. There is no black line.
  3. Figure 2 B: Perhaps use different colors (just a suggestion)
  4. Line 201: blue and violet traces in Figure 2 C. There is no violet trace.
  5. Line 235: path-clamp. Should be patch-clamp?
  6. Several diagrams differ in their size.

Re: We accept all these remarks. All required corrections are introduced in the text. 

Round 2

Reviewer 2 Report

The authors satisfactorily addressed some of my previous concerns. However, the following points of my first revision still need adjustments before paper acceptance.

- Remark 5: I don’t think the authors’ reply satisfactorily answered my question. The treatment with TG+ caffeine is not the matter of my question. Instead, It should be discussed why, upon PC treatment, SR and mitochondrial Ca2+ levels change almost immediately, whereas cytosolic Ca2+ concentration needs 5 min to experience significant elevation. This is relevant, because not all the Ca2+ released from SR is taken up by mitochondria and a rise in cytosolic Ca2+ concentration should be observed whenever the cation is released from SR. Are traces in fig. 1a (or in fig. 4a) representative?

- Remark 7: “n was added for all panels”…That’s not the case in fig. 5. In my version, I don’t see n from panels A to E.

- Remark 9: Provided that I would have preferred to see these experiments in the revised version (they are quite simple for this team and just a few time would have been necessary), I think that at least in the discussion it should be mentioned that the available data might be also compatible with Ca2+-independent, but phospholipase-dependent mechanisms. I’m convinced that Ca2+ is important, but these data do not unambiguously demonstrate it.

Author Response

Remark 5: I don’t think the authors’ reply satisfactorily answered my question. The treatment with TG+ caffeine is not the matter of my question. Instead, It should be discussed why, upon PC treatment, SR and mitochondrial Ca2+ levels change almost immediately, whereas cytosolic Ca2+ concentration needs 5 min to experience significant elevation. This is relevant, because not all the Ca2+ released from SR is taken up by mitochondria and a rise in cytosolic Ca2+ concentration should be observed whenever the cation is released from SR. Are traces in fig. 1a (or in fig. 4a) representative?

Re: The signals of Fig.1A were registered with confocal microscope in XT mode at the speed 400 lines per second, the acquisition conditions which require relatively high laser intensity. Here, laser-induced effect on [Ca2+]i dynamics, i.e. sparks (but not spots) evoked by ROS, may overlap with PC effect. And, integral signal registered from the area will show slow smooth rise of [Ca2+]i within several minutes. This is the effect shown in control experiments without acylcarnitines application. Results presented at Figs. 1C-F, 2, 4, 7 were obtained with conventional epifluorescence microscope, equipped with HBO lamp. And a minimal required excitation light intensity was used without any effect on [Ca2+]i in control experiments. Here, we register slow smooth rise of [Ca2+]i only at weak mitochondrial Ca2+ buffering (Ruthenium Red, low Pi, inhibition of OXPHOS, Figs. 2B-D). In control experiment (50 uM PC, black line at Fig. 4D), [Ca2+]i was stabilized within lag period at minimal level due to SR-mitochondrial interplay, while in the experiment with inhibited OXPHOS even 5-10 uM PC evoked slow rise of [Ca2+]i (green line).

As for Fig. 4A, both traces are representative, each describing separate experiment. The panel was obtained by overlaying 2 curves. MagFluo-4 was applied to monitor the alterations in [Ca2+]SR. This dye chelates both Ca2+ and Mg2+ with a Kd for Mg2+ of 4.7 mM and a Kd for Ca2+ of 22 µM and can be used as a low-affinity Ca2+ indicator. Therefore, we cannot determine absolute value or relative alteration in SR Ca2+ evoked by 5 uM PC to evaluate its impact quantitatively Here, we may suggest that the effect of PC on bulk free [Ca2+]I is too weak to be registered within the lag period when efficient SR-mitochondria interplay take place.

Remark 7: “n was added for all panels”…That’s not the case in fig. 5. In my version, I don’t see n from panels A to E.

Re: Thank you. We incorporated in the legend of Fig. 5 this: Panels A to E each display one of 3 independent experiments.

Remark 9: Provided that I would have preferred to see these experiments in the revised version (they are quite simple for this team and just a few time would have been necessary), I think that at least in the discussion it should be mentioned that the available data might be also compatible with Ca2+-independent, but phospholipase-dependent mechanisms. I’m convinced that Ca2+ is important, but these data do not unambiguously demonstrate it.

Re: We appreciate and recognize that this is very important question. At the moment, more questions than answers. For example: What are the mechanisms preventing mitochondrial Ca2+ overload and mPTP opening and activation of calpains in contracted oval shape cells? As for Ca2+-independent, but phospholipase-dependent mechanisms of cell death, we also consider calcium as number one player. The experiment presented at Fig. 2A (green line) indicated that PC cannot induce cell death in the absence of Ca2+, when the cells were incubated in Ca2+-free media for 20 min. While, after 2 min preincubation in Ca2+-free media PC evoked Ca2+-overload and cell death (blue line). We suppose that PC-calcium-arachidonic acid interplay may underlie cell death. Arachdonic acid is far more toxic than PC. We fully agree that this issue should have been considered in the Discussion and did this in item 3.7.

Reviewer 3 Report

The authors revised their manuscript carefully and tried to included all suggestions. 

Two comments:

  • Figure legend 1B: Do you really used 20 M PC? In the text before you stated 20 µM
  • There is still Fluo3 in Figure C? Is it 4 or 3?

Author Response

To Reviewer 3

The authors revised their manuscript carefully and tried to included all suggestions. 

Two comments:

  • Figure legend 1B: Do you really used 20 M PC? In the text before you stated 20 µM
  • There is still Fluo3 in Figure C? Is it 4 or 3?

Re: We appreciate and have made the appropriate corrections by substituting 20 M by 20 µM, and Fluo-3 by Fluo-4.